# BACKDIFF: A DIFFUSION MODEL FOR GENERALIZED TRANSFERABLE PROTEIN BACKMAPPING

## ABSTRACT

Coarse-grained (CG) models play a crucial role in the study of protein structures, protein thermodynamic properties, and protein conformation dynamics. Due to the information loss in the coarse-graining process, backmapping from CG to all-atom configurations is essential in many protein design and drug discovery applications when detailed atomic representations are needed for in-depth studies. Despite recent progress in data-driven backmapping approaches, devising a backmapping method that can be universally applied across various CG models and proteins remains unresolved. In this work, we propose BackDiff, a new generative model designed to achieve generalization and reliability in the protein backmapping problem. BackDiff leverages the conditional score-based diffusion model with geometric representations. Since different CG models can contain different coarse-grained sites which include selected atoms (CG atoms) and simple CG auxiliary functions of atomistic coordinates (CG auxiliary variables), we design a self-supervised training framework to adapt to different CG atoms, and constrain the diffusion sampling paths with arbitrary CG auxiliary variables as conditions. Our method facilitates end-to-end training and allows efficient sampling across different proteins and diverse CG models without the need for re-training. Comprehensive experiments over multiple popular CG models demonstrate BackDiff's superior performance to existing state-of-the-art approaches, and generalization and flexibility that these approaches cannot achieve. A pretrained BackDiff model can offer a convenient yet reliable plug-and-play solution for protein researchers, enabling them to investigate further from their own CG models.

## 1 INTRODUCTION

All-atom molecular dynamics (MD) simulations provide detailed insights into the atomic-level interactions and dynamics of proteins (Ciccotti et al. (2014)). However, the computational cost associated with these simulations is substantial, especially when considering large biological systems or long simulation timescales (Shaw et al. (2010)). The intricacies of atomic interactions necessitate small time steps and slow atomic force evaluations, making it challenging to model slow biological processes, such as protein folding, protein-protein interaction, and protein aggregation. Coarse-grained (CG) simulations emerge as an essential tool to address these challenges (Kmiecik et al. (2016); Marrink et al. (2007); Rudd & Broughton (1998)). By simplifying and grouping atoms into larger interaction units, CG models significantly reduce degrees of freedom, allowing for larger simulation length- and time-scales. A CG representation can be classified into two components: CG atoms and CG auxiliary variables. CG atoms denote those that are direct all-atom particles, meaning that each CG atom corresponds to an atom in the all-atom configuration. On the other hand, CG auxiliary variables function as mathematical representations to capture aggregate properties of groups of atoms. While many traditional physics-based CG models use the side chain center of mass (COM) as their CG auxiliary variables, recent CG models can adapt optimized nonlinear (Diggins IV et al. (2018)) or data-driven CG auxiliary variables (Fu et al. (2022)) to give a more comprehensive description of proteins.

However, the coarse-graining process sacrifices atomic-level precision, which, in many cases, is essential for a comprehensive understanding of the biomolecular system, such as the drug binding process. Thus, retrieving all-atom configurations by backmapping the CG configurations is important for a more detailed and accurate modeling of proteins. Traditional simulation-based

backmapping methods, which perform by equilibrating configurations through MC or MD simulation (Badaczewska-Dawid et al. (2020); Vickery & Stansfeld (2021); Liu et al. (2008)), are computationally expensive and highly intricate, thus diminishing the value of CG simulations. In response, recent data-driven methods (Li et al. (2020); Louison et al. (2021); An & Deshmukh (2020)) employ generative models for more efficient and accurate protein backmapping. These models learn the probability distribution of all-atom configurations conditioned on CG structures, and can efficiently sample from the distribution. Yang & Gómez-Bombarelli (2023) extends the generative backmapping by aiming for transferability in protein spaces. While these methods have shown promising results for backmapping various proteins, they often train and sample under a single, predefined CG model. Wang et al. (2022) illustrates that the method can be adapted to CG models with different resolutions. However, model retraining is needed for each adjustment.

In this study, we introduce BackDiff, a deep generative backmapping approach built upon conditional score-based diffusion models (Song et al. (2020)). BackDiff aims to achieve transferability across different proteins and generalization across CG methods. The high-level idea of BackDiff is to resolve the transferability of CG atoms at the training phase and CG auxiliary variables at the sampling phase. The primary objective of the training is to reconstruct the missing atoms by learning the probability distribution of these atoms conditioned on CG atoms, using conditional score-based diffusion models. The diffusion model gradually transitions the missing atoms from their original states to a noisy distribution via a forward diffusion process, and learns the reverse diffusion process, which recovers the target geometric distribution from the noise. In order to train a model transferable to multiple CG methods, we develop a self-supervised training method that semi-randomly selects CG atoms in each epoch during training. Due to the variability of CG auxiliary functions, it's infeasible to train a single model adaptable to all CG auxiliary variables. By harnessing the unique properties of the score-based diffusion model, we address this challenge through manifold constraint sampling (Chung et al. (2022)), allowing for flexibility across different CG auxiliary variables. The CG auxiliary variables act as a guiding constraint to the data manifold during the reverse diffusion process, ensuring that our sampled data remains within the generative boundaries defined by these CG auxiliary variables.

We employ BackDiff for extensive backmapping experiments across various popular CG models, all without the need for model retraining. Numerical evaluations demonstrate that BackDiff consistently delivers robust performance across diverse CG models, and commendable transferability across protein spaces, even when data is limited.

## 2 RELATED WORK

### 2.1 SCORE-BASED DIFFUSION MODELS

The score-based diffusion model perturbs data with original distribution $p_0(\mathbf{x})$ to noise with a diffusion process over a unit time horizon by a linear stochastic differential equation (SDE):

$$d\mathbf{x} = \mathbf{f}(\mathbf{x}, t)dt + g(t)d\mathbf{w}, \ t \in [0, T], \tag{1}$$

where $\mathbf{f}(\mathbf{x}, t), g(t)$ are chosen diffusion and drift functions and $\mathbf{w}$ denotes a standard Wiener process. With a sufficient amount of time steps, the prior distribution $p_T(\mathbf{x})$ approaches a simple Gaussian distribution. For any diffusion process in equation 1, it has a corresponding reverse-time SDE (Anderson (1982)):

$$d\mathbf{x} = [\mathbf{f}(\mathbf{x}, t) - g^2(t)\nabla_{\mathbf{x}_t} \log p_t(\mathbf{x}_t)]dt + g(t)d\bar{\mathbf{w}}, \tag{2}$$

with $\bar{\mathbf{w}}$ a standard Wiener process in the reverse-time. The trajectories of the reverse SDE (2) have the same marginal densities as the forward SDE (1). Thus, the reverse-time SDE (2) can gradually convert noise to data. The score-based diffusion model parameterizes the time-dependent score function $\nabla_{\mathbf{x}_t} \log p_t(\mathbf{x}_t)$ in the reverse SDE (2) with a neural network $\mathbf{s}_\theta(\mathbf{x}(t), t)$. The time-dependent score-based model $\mathbf{s}_\theta(\mathbf{x}(t), t)$ can be trained via minimizing the denoising score matching loss:

$$J(\theta) = \arg\min_\theta \mathbb{E}_t \left\{ \mathbb{E}_{\mathbf{x}(0)} \mathbb{E}_{\mathbf{x}(t)|\mathbf{x}(0)} \left[ \|\mathbf{s}_\theta(\mathbf{x}(t), t) - \nabla_{\mathbf{x}_t} \log p(\mathbf{x}(t) \mid \mathbf{x}(0))\|_2^2 \right] \right\}, \tag{3}$$

with $t$ uniformly sampled between $[0, T]$. To sample from the data distribution $p(\mathbf{x})$, we first draw a sample from the prior distribution $p(\mathbf{x}_T) \sim \mathcal{N}(\mathbf{0}, \boldsymbol{I})$, and then discretize and solve the reverse-time SDE with numerical methods, e.g. Euler-Maruyama discretization.

In this work, we consider the variance preserving (VP) form of the SDE in Denoising Diffusion Probabilistic Model (DDPM) (Ho et al. (2020)):

$$d\mathbf{x} = -\frac{1}{2}\beta(t)\mathbf{x}dt + \sqrt{\beta(t)}d\mathbf{w}, \tag{4}$$

with $\beta(t)$ representing the variance schedule. In a discretized setting of DDPM, we define $\beta_1, \beta_2, ..., \beta_T$ as the sequence of fixed variance schedule, $\alpha_t = 1 - \beta_t$ and $\bar{\alpha}_t = \prod_{i=1}^{t} \alpha_i$, then the forward diffusion process can be written as:

$$\mathbf{x}_t = \sqrt{\bar{\alpha}_t}\mathbf{x}_0 + \sqrt{1 - \bar{\alpha}_t}\mathbf{z}, \quad \mathbf{z} \sim \mathcal{N}(\mathbf{0}, \boldsymbol{I}). \tag{5}$$

## 2.2 CONDITIONAL SCORE-BASED DIFFUSION MODEL FOR IMPUTATION PROBLEMS

Let us consider a general missing value imputation problem: given a sample $\mathbf{x} \equiv \{\mathbf{x}_{\text{known}}, \mathbf{x}_{\text{omit}}\}$, where $\mathbf{x}_{\text{known}}$ represents observed values and $\mathbf{x}_{\text{omit}}$ represents missing values, and $\mathbf{x}_{\text{known}}, \mathbf{x}_{\text{omit}}$ can vary by samples, we want to recover $\mathbf{x}_{\text{omit}}$ with the conditional observed values $\mathbf{x}_{\text{known}}$. In the context of protein backmapping, $\mathbf{x}_{\text{known}}$ represents the atomic coordinates of CG atoms, while $\mathbf{x}_{\text{omit}}$ denotes the atomic coordinates to be recovered. Thus, the imputation problem can be formulated as learning the true conditional probability $p(\mathbf{x}_{\text{omit}}|\mathbf{x}_{\text{known}})$ with a parameterized distribution $p_\theta(\mathbf{x}_{\text{omit}}|\mathbf{x}_{\text{known}})$.

We can apply score-based diffusion model to the imputation problem by incorporating the conditional observed values into the reverse diffusion from equation 2:

$$d\mathbf{x}_{\text{omit}} = [\mathbf{f}(\mathbf{x}_{\text{omit}}, t) - g^2(t)\nabla_{\mathbf{x}_{\text{omit}_t}} \log p_t(\mathbf{x}_{\text{omit}_t}|\mathbf{x}_{\text{known}})]dt + g(t)d\bar{\mathbf{w}}. \tag{6}$$

## 2.3 MANIFOLD CONSTRAINT SAMPLING FOR INVERSE PROBLEMS

Consider a many-to-many mapping function $\mathcal{A} : \mathbf{X} \to \mathbf{Y}$. The inverse problem is to retrieve the distribution of $\mathbf{x} \in \mathbf{X}$, which can be multimodal, given a measurement $\mathbf{y} \in \mathbf{Y}$. In the protein backmapping problem, $\mathbf{y}$ corresponds to the CG auxiliary variables and $\mathbf{x}$ the atomic coordinates to recover. With the Bayes' rule:

$$p(\mathbf{x}|\mathbf{y}) = p(\mathbf{y}|\mathbf{x})p(\mathbf{x})/p(\mathbf{y}), \tag{7}$$

we can take $p(\mathbf{x})$ as the prior and sample from the posterior $p(\mathbf{x}|\mathbf{y})$. If we take the score-based diffusion model as the prior, we can use the reverse diffusion from equation 2 as the sampler from the posterior distribution as follows:

$$d\mathbf{x} = [\mathbf{f}(\mathbf{x}, t) - g^2(t)(\nabla_{\mathbf{x}_t} \log p_t(\mathbf{x}_t) + \nabla_{\mathbf{x}_t} \log p_t(\mathbf{y}|\mathbf{x}_t))]dt + g(t)d\bar{\mathbf{w}}, \tag{8}$$

using the Bayes' rule:

$$\nabla_{\mathbf{x}_t} \log p_t(\mathbf{x}_t|\mathbf{y}) = \nabla_{\mathbf{x}_t} \log p_t(\mathbf{x}_t) + \nabla_{\mathbf{x}_t} \log p_t(\mathbf{y}|\mathbf{x}_t). \tag{9}$$

By estimating the score function $\nabla_{\mathbf{x}_t} \log p_t(\mathbf{x}_t)$ with a trained score model $\mathbf{s}_\theta$ and computing the likelihood $\nabla_{\mathbf{x}_t} \log p_t(\mathbf{y}|\mathbf{x}_t)$, we can obtain the posterior likelihood $\nabla_{\mathbf{x}_t} \log p_t(\mathbf{x}_t|\mathbf{y})$. However, computing $\nabla_{\mathbf{x}_t} \log p_t(\mathbf{y}|\mathbf{x}_t)$ in a closed-form is difficult since there is not a clear dependence between $\mathbf{y}$ and $\mathbf{x}_t$.

Observing that $\mathbf{y}$ and $\mathbf{x}_t$ are independently conditioned on $\mathbf{x}_0$, we can factorize $p(\mathbf{y}|\mathbf{x}_t)$ as:

$$p(\mathbf{y}|\mathbf{x}_t) = \int p(\mathbf{y}|\mathbf{x}_0, \mathbf{x}_t)p(\mathbf{x}_0|\mathbf{x}_t)d\mathbf{x}_0 = \int p(\mathbf{y}|\mathbf{x}_0)p(\mathbf{x}_0|\mathbf{x}_t)d\mathbf{x}_0, \tag{10}$$

which interprets the conditional probability as:

$$p(\mathbf{y} \mid \mathbf{x}_t) = \mathbb{E}_{\mathbf{x}_0 \sim p(\mathbf{x}_0|\mathbf{x}_t)} [p(\mathbf{y} \mid \mathbf{x}_0)]. \tag{11}$$

This further implies that we can approximate the conditional probability as:

$$p(\mathbf{y} \mid \mathbf{x}_t) \simeq p(\mathbf{y} \mid \hat{\mathbf{x}}_0), \tag{12}$$

where $\hat{\mathbf{x}}_0 = \mathbb{E}[\mathbf{x}_0 \mid \mathbf{x}_t]$.

Chung et al. (2022) proves that for DDPM, $p(\mathbf{x}_0|\mathbf{x}_t)$ has a unique posterior mean as:

$$\hat{\mathbf{x}}_0 = \frac{1}{\sqrt{\bar{\alpha}(t)}}(\mathbf{x}_t + (1 - \bar{\alpha}(t))\nabla_{\mathbf{x}_t} \log p_t(\mathbf{x}_t)). \tag{13}$$

The conditional probability $p(\mathbf{y}|\mathbf{x}_0)$ is a Dirac delta function $p(\mathbf{y}|\mathbf{x}_0) = \delta(\mathbf{y} - \mathcal{A}(\mathbf{x}_0))$. In practice, we replace the multidimensional $\delta$ function with a multidimensional Gaussian of small variance, which regularizes the strict constraints $\mathcal{A}(\mathbf{x}_0) = \mathbf{y}$ with a tight restraint:

$$p\left(\mathbf{y} \mid \mathbf{x}_0\right) = \frac{1}{\sqrt{(2\pi)^n \sigma^{2n}}} \exp\left[-\frac{\|\mathbf{y} - \mathcal{A}(\mathbf{x}_0)\|_2^2}{2\sigma^2}\right], \tag{14}$$

where $n$ is the dimension of $\mathbf{y}$ and $\sigma$ is the standard deviation. Taking the approximation from equation 12, we can get:

$$\nabla_{\mathbf{x}_t} \log p\left(\mathbf{y} \mid \mathbf{x}_t\right) \simeq \nabla_{\mathbf{x}_t} \log p\left(\mathbf{y} \mid \hat{\mathbf{x}}_0\right) = -\frac{1}{\sigma^2} \nabla_{\mathbf{x}_t} \|\mathbf{y} - \mathcal{A}(\hat{\mathbf{x}}_0)\|_2^2. \tag{15}$$

Finally, by estimating $\nabla_{\mathbf{x}_t} \log p_t(\mathbf{x}_t)$ with a neural network $\mathbf{s}_\theta(\mathbf{x}_t, t)$, we can formulate the conditional reverse diffusion of DDPM modified from equation 8 as:

$$d\mathbf{x} = [\mathbf{f}(\mathbf{x}, t) - g^2(t)(\mathbf{s}_\theta(\mathbf{x}_t, t) - \zeta \nabla_{\mathbf{x}_t} \|\mathbf{y} - \mathcal{A}(\hat{\mathbf{x}}_0)\|_2^2]dt + g(t)d\bar{\mathbf{w}}, \tag{16}$$

with $\hat{\mathbf{x}}_0$ expressed as in equation 13, and $\zeta = \frac{1}{\sigma^2}$ is the correction weight.

## 3 PRELIMINARY

### 3.1 NOTATIONS AND PROBLEM DEFINITION

**Notations.** In this paper, each protein with $N$ heavy atoms is represented as an undirected graph $\mathcal{G} = \langle \mathcal{V}, \mathcal{E} \rangle$, where $\mathcal{V} = \{v_i\}_{i=1}^N$ is the set of nodes representing heavy atoms and $\mathcal{E} = \{e_{ij}|(i,j) \subseteq |\mathcal{V}| \times |\mathcal{V}|\}$ is the set of edges representing inter-atomic bonds and nonbonded interactions. An all-atom configuration of the protein can be represented as $\mathcal{C} = [\boldsymbol{c}_1, \boldsymbol{c}_2, \cdots, \boldsymbol{c}_N] \in \mathbb{R}^{N \times 3}$, with $\boldsymbol{c}_i$ the Cartesian coordinates of $i$-th heavy atom. A CG model defines a CG mapping function $\xi$: $\mathcal{R} = \xi(\mathcal{C})$, that transforms the all-atom coordinate representation $\mathcal{C}$ to a lower-dimensional CG representation $\mathcal{R} \in \mathbb{R}^n$ ($n < 3N$). We further denote $\mathcal{R} \equiv \{\mathcal{R}_{\text{atm}}, \mathcal{R}_{\text{aux}}\}$, where $\mathcal{R}_{\text{atm}}$ represents CG atoms and $\mathcal{R}_{\text{aux}}$ are CG auxiliary variables.

**Problem Definition.** Given a protein graph $\mathcal{G}$ and a coarse-grained configuration $\mathcal{R}$, the task of protein backmapping is to learn and efficiently sample from $p(\mathcal{C}|\mathcal{R}, \mathcal{G})$. This will allow us to conduct CG MD simulations to longer time- and length- scales for any protein with a CG method chosen at will, and recover the lost information by sampling from the posterior distribution, without the need for retraining. In this work, we only require that the atomic coordinates of all alpha carbons ($C_\alpha$) are included in CG representations $\mathcal{R}$.

## 4 BACKDIFF METHOD

In this section, we elaborate on the proposed Backdiff framework. On the high level, Backdiff addresses the transferability of $\mathcal{R}_{\text{atm}}$ and $\mathcal{R}_{\text{aux}}$ distinctly, resolving them in different components of the work. During training, we develop a diffusion-based generative model to learn the distribution $p(\mathcal{C}|\mathcal{R}_{\text{atm}}, \mathcal{G})$. We approach this training by viewing it as a missing-node-imputation problem, and devise a self-supervised training strategy that can accommodate a wide range of missing-node combinations. During the sampling procedure, we enforce the condition of $\mathcal{R}_{\text{aux}}$ by incorporating a correction term through the reverse diffusion process. This ensures accurate sampling from the distribution $p(\mathcal{C}|\mathcal{R}_{\text{atm}}, \mathcal{R}_{\text{aux}}, \mathcal{G})$. Finally, we apply the same manifold constraint sampling technique on bond lengths and bond angles to avoid generating unrealistic protein configurations. In Sec. 4.1 and Sec. 4.2, we present a description of the training process and the self-supervised training strategy. In Sec. 4.3, we explain how BackDiff avoids dealing with equivariance. Finally, we show how to utilize manifold constraint sampling to adapt to arbitrary CG auxiliary variables $\mathcal{R}_{\text{aux}}$ and to enforce bond lengths and bond angles in Sec. 4.4 and Sec. 4.5. The high-level schematic of the sampling process is shown in Fig. 1. An equivariant Graph Neural Network is used in this paper to parameterize the score-based diffusion model. We elaborate on details of the GNN architecture used in Appendix G.1.

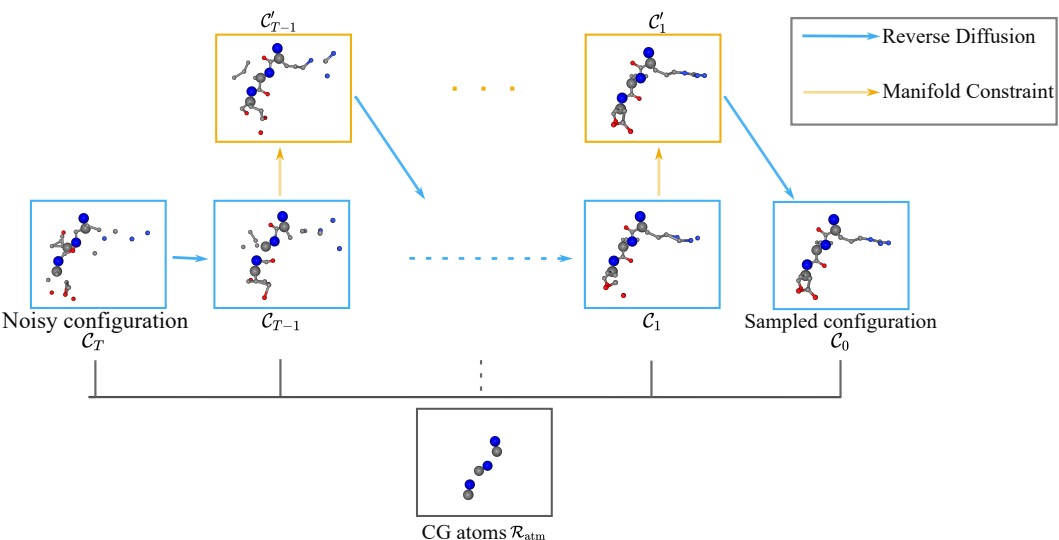

Figure 1: The sampling process of BackDiff. The reverse diffusion process gradually converts the noisy configuration into the plausible configuration, conditioned on CG atoms $\mathcal{R}_{\text{atm}}$. In each diffusion step, the configuration is "corrected" with auxiliary variables, bond lengths and bond angles as manifold constraints.

## 4.1 TRAINING FORMULATION

Let us denote the target all-atom configuration as $\mathcal{C} \equiv \{\mathcal{C}_{\text{omit}}, \mathcal{R}_{\text{atm}}\}$, with $\mathcal{C}_{\text{omit}}$ denoting the Cartesian coordinates of atoms omitted during the coarse-graining process. We further denote $\mathcal{D}$ the displacement of omitted atoms from the $C_\alpha$ of their corresponding residues. Since we require the atomic coordinates of $C_\alpha$ to be incorporated in the CG conditions, we can observe that $p(\mathcal{C}|\mathcal{R}_{\text{atm}}, \mathcal{G}) = p(\mathcal{C}_{\text{omit}}|\mathcal{R}_{\text{atm}}, \mathcal{G}) = p(\mathcal{D}|\mathcal{R}_{\text{atm}}, \mathcal{G})$. We choose $p(\mathcal{D}|\mathcal{R}_{\text{atm}}, \mathcal{G})$ as our learning target since compared to the Cartesian coordinates $\mathcal{C}_{\text{omit}}$, the displacement $\mathcal{D}$ spans a smaller data range and thus enhances training stability.

We model the conditional distribution $p(\mathcal{D}|\mathcal{R}_{\text{atm}}, \mathcal{G})$ using the score-based diffusion model with a modified reverse diffusion defined in equation 6. We define a parameterized conditional score function $\mathbf{s}_\theta : (\mathbf{D}_t \times \mathbb{R}|\mathbf{R}_{\text{atm}}) \rightarrow \mathbf{D}_t$ to approximate $\nabla_{\mathcal{D}_t} \log p_t(\mathcal{D}_t|\mathcal{R}_{\text{atm}})$. We follow the same training procedure for the unconditional score-based diffusion as described in Sec. 2.1: given the Cartesian coordinates of CG atoms $\mathcal{R}_{\text{atm}}$ and the displacement of omitted atoms from alpha carbons $\mathcal{D}$, we perturb the displacement $\mathcal{D}$ with DDPM forward diffusion process defined following equation 4:

$$\mathcal{D}_t = \sqrt{\bar{\alpha}_t}\mathcal{D}_0 + \sqrt{1 - \bar{\alpha}_t}\mathbf{z}, \quad \mathbf{z} \sim \mathcal{N}(\mathbf{0}, \boldsymbol{I}). \tag{17}$$

Next, we sample perturbed $\mathcal{D}$ and train $s_\theta$ by minimizing the loss function

$$J(\theta) = \arg\min_\theta \mathbb{E}_{t, \mathcal{D}(0), \mathcal{D}(t)|\mathcal{D}(0)} \left[ \left\| \mathbf{s}_\theta(\mathcal{D}(t), t|\mathcal{R}_{\text{atm}}) - \nabla_{\mathcal{D}(t)} \log p_{0t}(\mathcal{D}(t) \mid \mathcal{D}(0), \mathcal{R}_{\text{atm}}) \right\|_2^2 \right]. \tag{18}$$

Inspired by Tashiro et al. (2021), we develop a self-supervised learning framework for the backmapping problem. During each iteration of training, for each all-atom configuration, we choose a set of atoms as CG atoms $\mathcal{R}_{\text{atm}}$, following a semi-randomized strategy, and leave the rest of the atoms as omitted atoms $\mathcal{C}_{\text{omit}}$ and compute their displacements $\mathcal{D}$ from corresponding alpha carbons. During training, the choice of CG atoms will change from iteration to iteration.

## 4.2 CHOICE OF CG ATOMS IN SELF-SUPERVISED LEARNING

In this study, all $C_\alpha$ are enforced as CG atoms. For the rest of the atoms, we provide three strategies for choosing CG atoms during training. Each strategy can be chosen based on information known about the target proteins and CG methods.

(1) Random strategy: we randomly select a certain percentage of atoms as CG atoms. This strategy should be adopted if we do not know the common choices of CG atoms of CG models. The percentage is uniformly sampled from $[0\%, 100\%]$ to adapt to various CG resolutions.

(2) Semi-random strategy: for different types of atoms ($C, N$ on the backbone, $C_\beta, C_\gamma$ on the side chain, etc.), we assign different percentages to choose as CG atoms. This strategy should be adopted if we know the common choices of CG atoms but want to keep the diversity of training.

(3) Fix strategy: we choose a fixed set of atoms as CG atoms. This strategy is adopted if we want to train BackDiff with respect to a specific CG method.

More detailed descriptions of algorithms of methods (1) and (2) are given in Appendix C.2.

### 4.3 EQUIVARIANCE

Equivariance is a commonly used property in geometric deep learning (Satorras et al. (2021); Batzner et al. (2022); Maron et al. (2018)). A function $\phi : X \to Y$ is said to be equivariant w.r.t. a group $G$ if

$$\phi\left(T_g(\mathbf{x})\right) = S_g(\phi(\mathbf{x})), \tag{19}$$

where $T_g : X \to X$ and $S_g : Y \to Y$ are transformations of $g \in G$. In this work, we consider $G$ the SE(3) group, which is the group of rotation and translation.

**Proposition 1.** *Our training target $p(\mathcal{C}|\mathcal{R}_{atm}, \mathcal{G})$ is SE(3)-equivariant, i.e., $p(\mathcal{C}|\mathcal{R}_{atm}, \mathcal{G}) = p(T_g(\mathcal{C})|T_g(\mathcal{R}_{atm}), \mathcal{G})$, then for all diffusion time $t$, the time-dependent score function is SE(3)-equivariant:*

$$\begin{aligned}
\nabla_\mathcal{C} \log p_t(\mathcal{C}|\mathcal{R}_{atm}, \mathcal{G}) &= \nabla_\mathcal{C} \log p_t(T(\mathcal{C})|T(\mathcal{R}_{atm}), \mathcal{G}) \\
&= S(\nabla_\mathcal{C} \log p_t(S(\mathcal{C})|S(\mathcal{R}_{atm}), \mathcal{G}))
\end{aligned} \tag{20}$$

*for translation $T$ and rotation $S$.*

### 4.4 MANIFOLD CONSTRAINT SAMPLING ON CG AUXILIARY VARIABLES

Let us consider CG auxiliary variables $\mathcal{R}_{aux}$ obtained from a many-to-many mapping function $\xi_{aux}$:

$$\mathcal{R}_{aux} = \xi_{aux}(\mathcal{D}, \mathcal{R}_{atm}). \tag{21}$$

With a learned $\mathbf{s}_\theta(\mathcal{D}_t|\mathcal{R}_{atm}, \mathcal{G})$, our objective is to sample from $p_\mathcal{G}(\mathcal{D}|\mathcal{R}_{atm}, \mathcal{R}_{aux})$ for an arbitrary CG auxiliary function $\xi_{aux}$ with the score-based diffusion model. The sampling process, however, requires knowledge of $\nabla_{\mathcal{D}_t} \log p_\mathcal{G}(\mathcal{D}_t|\mathcal{R}_{atm}, \mathcal{R}_{aux})$. We can compute $\nabla_{\mathcal{D}_t} \log p_\mathcal{G}(\mathcal{D}_t|\mathcal{R}_{atm}, \mathcal{R}_{aux})$ from $\nabla_{\mathcal{D}_t} \log p_\mathcal{G}(\mathcal{D}_t|\mathcal{R}_{atm})$ using Baye's rule:

$$\begin{aligned}
\nabla_{\mathcal{D}_t} \log p_\mathcal{G}(\mathcal{D}_t|\mathcal{R}_{atm}, \mathcal{R}_{aux}) &= \nabla_{\mathcal{D}_t} \log p_\mathcal{G}(\mathcal{D}_t|\mathcal{R}_{atm}) \\
&+ \nabla_{\mathcal{D}_t} \log p_\mathcal{G}(\mathcal{R}_{aux}|\mathcal{R}_{atm}, \mathcal{D}_t).
\end{aligned} \tag{22}$$

This decomposition allows us to take $p_\mathcal{G}(\mathcal{D}_t|\mathcal{R}_{atm})$ as prior and sample from $p_\mathcal{G}(\mathcal{D}|\mathcal{R}_{atm}, \mathcal{R}_{aux})$ with the manifold constraint sampling technique. The first term $\nabla_{\mathcal{D}_t} \log p_\mathcal{G}(\mathcal{D}_t|\mathcal{R}_{atm})$ is estimated with $\mathbf{s}_\theta(\mathcal{D}_t|\mathcal{R}_{atm}, \mathcal{G})$, and the second term $\nabla_{\mathcal{D}_t} \log p_\mathcal{G}(\mathcal{R}_{aux}|\mathcal{R}_{atm}, \mathcal{D}_t)$ is estimated following equation 15:

$$\nabla_{\mathcal{D}_t} \log p_\mathcal{G}(\mathcal{R}_{aux}|\mathcal{R}_{atm}, \mathcal{D}_t) \simeq -\zeta \nabla_{\mathcal{D}_t} \left\| \mathcal{R}_{aux} - \xi_{\mathbf{aux}}\left(\hat{\mathcal{D}}_0, \mathcal{R}_{atm}\right) \right\|_2^2, \tag{23}$$

where $\hat{\mathcal{D}}_0$ is computed according to equation 13:

$$\hat{\mathcal{D}}_0 = \frac{1}{\sqrt{\bar{\alpha}(t)}}\left(\mathcal{D}_t + (1 - \bar{\alpha}(t))\nabla_{\mathcal{D}_t} \log p_\mathcal{G}(\mathcal{D}_t|\mathcal{R}_{atm})\right). \tag{24}$$

We provide the pseudo code of the sampling process in Appendix C.1.

### 4.5 MANIFOLD CONSTRAINT ON BOND LENGTHS AND ANGLES

In proteins, the bond lengths and bond angles exhibit only minor fluctuations due to the strong force constants of covalent bonds. This results in an ill-conditioned probability distribution for Cartesian coordinates. A diffusion model based on Cartesian coordinates faces challenges in accurately learning such a distribution, potentially leading to the generation of unrealistic configurations. In this work, we apply manifold constraints on bond lengths and bond angles in addition to CG auxiliary variables, as the posterior conditions.

## 5 EXPERIMENT

**Datasets** Following the recent protein backmapping work, we use the protein structural ensemble database PED (Lazar et al. (2021)) as our database. PED contains structural ensembles of 227 proteins, including intrinsically disordered protein (IDP). Among the 227 proteins, we choose 92 data computed from MD simulation or sampling methods for training and testing purposes.

**Evaluation** We conduct several experiments to demonstrate the flexibility, reliability, and transferability of BackDiff. We evaluate the performance of Backdiff on 3 popular CG models: UNRES model (Liwo et al. (2014)),Rosetta model (Das & Baker (2008), and MARTINI model (Souza et al. (2021)). The CG mapping protocol of each model is summarized in Table .5 in Appendix D. We perform both single- and multi-protein experiments, with single-protein experiments training and inference on one single protein, and multi-protein experiments training and inference on multiple proteins. Single-protein experiments are conducted on PED00011 (5926 frames) and PED00151 (9746 frames). We randomly split the training, validation, and testing datasets into PED00011 (3000 frames for training, 2826 frames for validation, 100 frames for testing), and PED00151 (4900 frames for training, 4746 frames for validation, and 100 frames for testing). For multi-protein experiments, we randomly select up to 500 frames for each protein from the dataset as the training dataset. For testing, we randomly select 100 frames other than the ones used in training for PED00011 and PED00151, and 45 frames other than the ones used in training for PED00055. In both single- and multi-protein experiments, we test BackDiff with fixed training strategies (CG-fixed) and BackDiff with semi-random training strategy (CG-transferable).

**Baselines** We choose GenZProt (Yang & Gómez-Bombarelli (2023)) and modified Torsional Diffusion (TD) (Jing et al. (2022)) as the state-of-the-art baselines. Since GenZProt and Torsional Diffusion utilize internal coordinates (torsion angles) as training objectives and adapting them to multiple CG methods can be ill-defined, we conduct single- and multi-protein experiments with fixed CG methods for the two baseline models.

**Evaluation Metrics** Since backmapping generates multiple configurations ($\mathcal{C}_{\text{gen}}$) from one CG configuration, a good protein backmapping model should be able to generate some samples that match the original all-atom configuration ($\mathcal{C}_{\text{ref}}$) (accuracy), consist of new configurations (diversity), and are physically realistic. For the accuracy metrics, we identify one generated sample ($\mathcal{C}_{\text{min}}$) with the minimum Root Mean Squared Distance (RMSD$_{\text{min}}$) w.r.t $\mathcal{C}_{\text{ref}}$, and compute the Mean Square Error (MSE) of $\mathcal{C}_{\text{min}}$'s sidechain COMs from $\mathcal{C}_{\text{ref}}$ (SCMSE$_{\text{min}}$). We report the mean and standard deviation of the RMSD$_{\text{min}}$ and SCMSE$_{\text{min}}$ across all testing frames. A lower RMSD$_{\text{min}}$ and SCMSE$_{\text{min}}$ indicate the model's stronger capacity to find $\mathcal{C}_{\text{ref}}$ as one representitave sample. For the diversity metric, we evaluate the generative diversity score (DIV) of $\mathcal{C}_{\text{gen}}$ and $\mathcal{C}_{\text{ref}}$, as suggested in Jones et al. (2023): DIV($\mathcal{C}_{\text{gen}}, \mathcal{C}_{\text{ref}}$). Full definitions of DIV is provided in Appendix F. A lower DIV suggests that the model can generate diverse $\mathcal{C}_{\text{gen}}$. Finally, we use steric clash ratio (SCR) to evaluate whether a model can generate physically realistic samples. SCR is defined following the metric in GenZProt: the ratio of steric clash occurrence in all atom-atom pairs within 5.0 Å, where the steric clash is defined as an atom-atom pair with a distance smaller than 1.2 Å.

We also perform ablation studies to assess the impact of constraining bond lengths and bond angles during BackDiff's sampling. This evaluation uses the Mean Absolute Error (MAE) to compare bond lengths and angles between the ground truth and generated samples. Since GenZProt and Torsional Diffusion construct all-atom configurations from internal coordinates (bond lengths, bond angles, and torsion angles), and inherently prevent unrealistic bond lengths and angles, we exclude their errors from the report.

**Results and discussions** The evaluation metric results on UNRES CG model are summarized in Table 1 and Table 2. As shown in the tables, BackDiff consistently outperforms the state-of-the-art ML models in both single- and multi-protein experiments, and is capable of generating all-atom configurations of higher accuracy, diversity and physical significance. Notably, even when BackDiff is trained for transferability across various CG methods, it maintains performance comparable to training with a fixed CG method. This underscores BackDiff's robust generalization and its reliability in adapting to diverse CG methods. A closer look at the sampled structures, as visualized in Figure 2, reveals that BackDiff more accurately recovers local structures.

| | Method | PED00011 | PED00151 |
|---|---|---|---|
| $\text{RMSD}_{\text{min}}$ (Å) | **BackDiff (fixed)** | **0.415(0.107)** | **0.526(0.125)** |
| | BackDiff (trans) | 0.598(0.112) | 0.663(0.182) |
| | GenZProt | 1.392(0.276) | 1.246(0.257) |
| | TD | 1.035(0.158) | 1.253(0.332) |
| SCR (%) | **BackDiff (fixed)** | **0.100(0.035)** | **0.105(0.063)** |
| | BackDiff (trans) | 0.216(0.178) | 0.320(0.157) |
| | GenZProt | 0.408(0.392) | 0.647(0.384) |
| | TD | 0.356(0.303) | 0.452(0.187) |
| $\text{SCMSE}_{\text{min}}$ (Å$^2$) | **BackDiff (fixed)** | **0.045(0.008)** | **0.049(0.021)** |
| | BackDiff (trans) | 0.061(0.010) | 0.104(0.038) |
| | GenZProt | 1.225(0.121) | 1.340(0.182) |
| | TD | 1.134(0.125) | 1.271(0.158) |
| DIV (Å) | **BackDiff (fixed)** | **0.045(0.027)** | **0.072(0.034)** |
| | BackDiff (trans) | 0.144(0.045) | 0.201(0.032) |
| | GenZProt | 0.453(0.241) | 0.527(0.185) |
| | TD | 0.128(0.064) | 0.146(0.049) |

Table 1: Results on single-protein experiments backmapping from UNRES CG model. The method labeled "BackDiff (trans)" is CG-transferable, while the other three are CG-fixed. We report the mean and standard deviation for 100 generated samples.

| | Method | PED00011 | PED00055 | PED00151 |
|---|---|---|---|---|
| $\text{RMSD}_{\text{min}}$(Å) | **BackDiff (fixed)** | **0.652(0.214)** | 1.690(0.372) | **1.292(0.160)** |
| | BackDiff (trans) | 0.708(0.188) | **1.340(0.237)** | 1.435(0.226) |
| | GenZProt | 2.337(0.466) | 2.741(0.515) | 2.634(0.353) |
| | TD | 1.714(0.385) | 2.282(0.400) | 1.634(0.282) |
| SCR (%) | **BackDiff (fixed)** | **0.626(0.482)** | 0.829(0.546) | **0.463(0.268)** |
| | BackDiff (trans) | 0.918(0.609) | **0.786(0.335)** | 0.820(0.316) |
| | GenZProt | 2.347(1.289) | 2.477(0.448) | 1.545(0.602) |
| | TD | 0.983(0.476) | 1.584(0.501) | 0.620(0.320) |
| $\text{SCMSE}_{\text{min}}$ (Å$^2$) | **BackDiff (fixed)** | **0.076(0.012)** | 0.103(0.026) | **0.100(0.021)** |
| | BackDiff (trans) | 0.082(0.027) | **0.088(0.015)** | 0.123(0.040) |
| | GenZProt | 1.951(0.327) | 1.784(0.402) | 1.869(0.330) |
| | TD | 1.320(0.282) | 1.195(0.318) | 1.717(0.397) |
| DIV | BackDiff (fixed) | 0.155(0.069) | 0.276(0.109) | 0.213(0.087) |
| | **BackDiff (trans)** | **0.079(0.052)** | **0.143(0.067)** | **0.122(0.060)** |
| | GenZProt | 0.636(0.132) | 0.662(0.147) | 0.612(0.143) |
| | TD | 0.179(0.066) | 0.252(0.086) | 0.201(0.075) |

Table 2: Results on multi-protein experiments backmapping from UNRES CG model.

As noted earlier, one limitation of the Cartesian-coordinate-based diffusion model is its inability to consistently produce realistic bond lengths and bend angles, given that these typically fluctuate within narrow ranges. In Table 3, we present the MAE for both bond lengths and bond angles, illustrating the benefits of constraining the sampling diffusion path using these parameters as posterior conditions. The results clearly indicate that manifold constraint sampling substantially reduces the errors in bond lengths and angles, enhancing the model's overall performance.

| | Method | PED00011 | PED00055 | PED00151 |
|---|---|---|---|---|
| Bond length MAE (Å) | **BackDiff (cons)** | $< 0.001$ | $< 0.001$ | $< 0.001$ |
| | BackDiff (plain) | 0.542(0.047) | 0.496(0.055) | 0.332(0.032) |
| Bond angle MAE | **BackDiff (cons)** | **0.167(0.095)** | **0.106(0.088)** | **0.124(0.097)** |
| | BackDiff (plain) | 0.333(0.071) | 0.245(0.070) | 0.251(0.082) |
| SCR (%) | **BackDiff (cons)** | **0.918(0.609)** | **0.786(0.335)** | **0.820(0.316)** |
| | BackDiff (plain) | 2.884(0.813) | 2.507(0.654) | 2.301(0.344) |

Table 3: Ablation study on the bond lengths and bond angles manifold constraint sampling. Comparing configurations generated with manifold constraint sampling (BackDiff (cons)) and without the manifold constraint sampling (BackDiff (plain)) in multi-protein experiments backmapping from UNRES CG model. Both tests use the same trained CG-transferable BackDiff model.

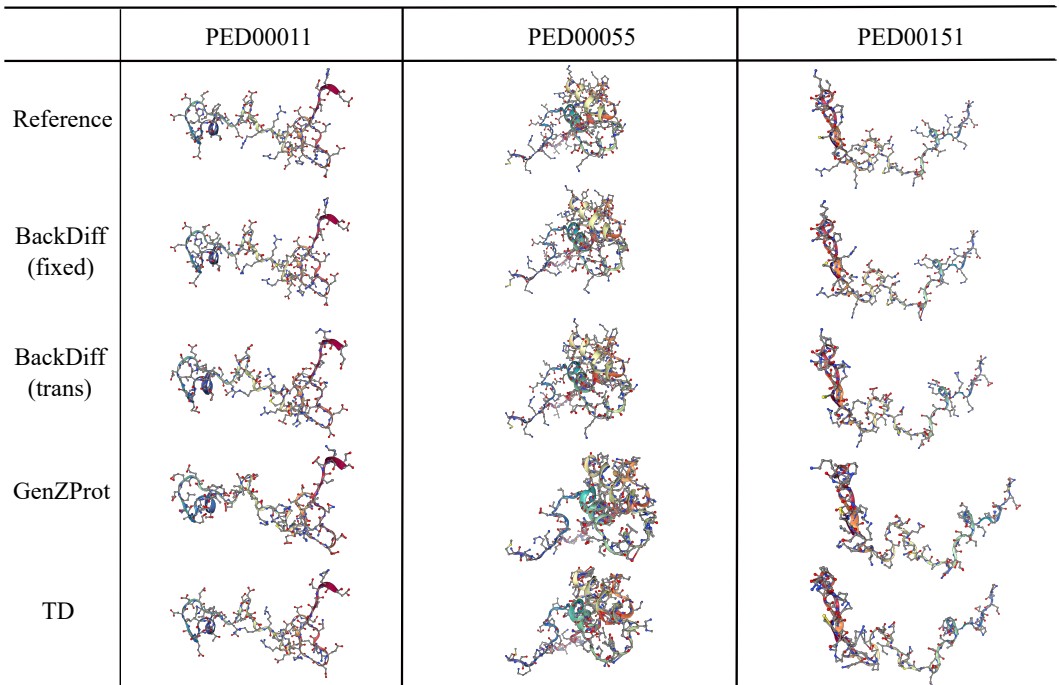

Figure 2: Visualization of all-atom configurations sampled from different methods in multi-protein experiments backmapping from UNRES CG model.

## 6 CONCLUSION

In this work, we propose BackDiff, a generative model for recovering proteins' all-atom structures from coarse-grained simulations. BackDiff combines a self-supervised score-based diffusion model with manifold constraint sampling to adapt to different CG models and utilizes geometric representations to achieve transferability across different proteins. Our rigorous experiments across various prominent CG models underscore BackDiff's exceptional performance and unparalleled adaptability. Looking ahead, we aim to improve the sampling efficiency of the diffusion model, refine the manifold constraint sampling process, integrate a more robust dataset to further enhance the model's capabilities, and expand our experimental scope to include recent CG models with data-driven mapping protocols.

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

# A PROOF

## A.1 PROOF OF EQUATION 22

Let us rewrite $\nabla_{\mathcal{D}_t} \log p_t(\mathcal{D}_t|\mathcal{R}_{\mathrm{atm}}, \mathcal{G})$ and $\nabla_{\mathcal{D}_t} \log p_t(\mathcal{D}_t|\mathcal{R}_{\mathrm{atm}}, \mathcal{R}_{\mathrm{aux}}, \mathcal{G})$ using Baye's rule:

$$\nabla_{\mathcal{D}_t} \log p_{\mathcal{G}}(\mathcal{D}|\mathcal{R}_{\mathrm{atm}}) = \nabla_{\mathcal{D}_t} \log p_{\mathcal{G}}(\mathcal{R}_{\mathrm{atm}}|\mathcal{D}_t) + \nabla_{\mathcal{D}_t} \log p_{\mathcal{G}}(\mathcal{D}_t)$$

$$\begin{aligned}
\nabla_{\mathcal{D}_t} \log p_{\mathcal{G}}(\mathcal{D}_t|\{\mathcal{R}_{\mathrm{atm}}, \mathcal{R}_{\mathrm{aux}}\}) &= \nabla_{\mathcal{D}_t} \log p_{\mathcal{G}}(\{\mathcal{R}_{\mathrm{atm}}, \mathcal{R}_{\mathrm{aux}}\}|\mathcal{D}_t) + \nabla_{\mathcal{D}_t} \log p_{\mathcal{G}}(\mathcal{D}_t) \\
&= \nabla_{\mathcal{D}_t} \log p_{\mathcal{G}}(\mathcal{R}_{\mathrm{atm}}|\mathcal{D}_t) \\
&\quad + \nabla_{\mathcal{D}_t} \log p_{\mathcal{G}}(\mathcal{R}_{\mathrm{aux}}|\{\mathcal{R}_{\mathrm{atm}}, \mathcal{D}_t\}) \\
&\quad + \nabla_{\mathcal{D}_t} \log p_{\mathcal{G}}(\mathcal{D}_t),
\end{aligned} \tag{25}$$

and we complete the proof.

## A.2 PROOF OF PROPOSITION 1.

**Proposition 1.** *Our training target $p(\mathcal{C}|\mathcal{R}_{atm}, \mathcal{G})$ is SE(3)-equivariant, i.e., $p(\mathcal{C}|\mathcal{R}_{atm}, \mathcal{G}) = p(T_g(\mathcal{C})|T_g(\mathcal{R}_{atm}), \mathcal{G})$, then for all diffusion time $t$, the time-dependent score function is SE(3)-equivariant:*

$$\begin{aligned}
\nabla_{\mathcal{C}} \log p_t(\mathcal{C}|\mathcal{R}_{atm}, \mathcal{G}) &= \nabla_{\mathcal{C}} \log p_t(T(\mathcal{C})|T(\mathcal{R}_{atm}), \mathcal{G}) \\
&= S(\nabla_{\mathcal{C}} \log p_t(S(\mathcal{C})|S(\mathcal{R}_{atm}), \mathcal{G}))
\end{aligned} \tag{26}$$

*for translation $T$ and rotation $S$.*

*Proof.* In VP-SDE, the perturbation kernel can be written as:

$$p_{t|0}(\mathcal{C}(t) \mid \mathcal{C}(0)) = \mathcal{N}\left(\mathcal{C}(t); \mathcal{C}(0)e^{-\frac{1}{2}\int_0^t \beta(s)\mathrm{d}s}, \mathbf{I} - \mathbf{I}e^{-\int_0^t \beta(s)\mathrm{d}s}\right), \tag{27}$$

which is SE(3) equivariant. We can link the perturbation kernel under translation and rotation:

$$\begin{aligned}
p_t(\mathcal{C}|\mathcal{R}_{\mathrm{atm}}, \mathcal{G}) &= \int p_0(\mathcal{C}'|\mathcal{R}_{\mathrm{atm}}, \mathcal{G}) p_{t|0}(\mathcal{C}|\mathcal{C}') d\mathcal{C}' \\
&= \int p_0(T_g(\mathcal{C}')|T_g(\mathcal{R}_{\mathrm{atm}}), \mathcal{G}) p_{t|0}(T_g(\mathcal{C})|T_g(\mathcal{C}')) dT_g(\mathcal{C}') \\
&= p_t(T_g(\mathcal{C})|T_g(\mathcal{R}_{\mathrm{atm}}), \mathcal{G}).
\end{aligned} \tag{28}$$

For $T$ being translational transformation, we have:

$$\begin{aligned}
\nabla_{\mathcal{C}} \log p_t(\mathcal{C}|\mathcal{R}_{\mathrm{atm}}, \mathcal{G}) &= \nabla_{\mathcal{C}} \log p_t(T(\mathcal{C})|T(\mathcal{R}_{\mathrm{atm}}), \mathcal{G}) \\
&= \frac{\partial T(\mathcal{C})}{\partial \mathcal{C}} \nabla_{T(\mathcal{C})} \log p_t(T(\mathcal{C})|T(\mathcal{R}_{\mathrm{atm}}), \mathcal{G}) \\
&= \nabla_{T(\mathcal{C})} \log p_t(T(\mathcal{C})|T(\mathcal{R}_{\mathrm{atm}}), \mathcal{G}).
\end{aligned} \tag{29}$$

Similarly, for $S$ being rotational transformation, we have

$$\begin{aligned}
\nabla_{\mathcal{C}} \log p_t(\mathcal{C}|\mathcal{R}_{\mathrm{atm}}, \mathcal{G}) &= \nabla_{\mathcal{C}} \log p_t(S(\mathcal{C})|S(\mathcal{R}_{\mathrm{atm}}), \mathcal{G}) \\
&= \frac{\partial S(\mathcal{C})}{\partial \mathcal{C}} \nabla_{S(\mathcal{C})} \log p_t(S(\mathcal{C})|S(\mathcal{R}_{\mathrm{atm}}), \mathcal{G}) \\
&= S(\nabla_{S(\mathcal{C})} \log p_t(S(\mathcal{C})|S(\mathcal{R}_{\mathrm{atm}}), \mathcal{G})),
\end{aligned} \tag{30}$$

and we complete the proof. $\square$

# B DETAILS OF DENOISING DIFFUSION PROBABILISTIC MODELS AND SCORE-BASED DIFFUSION MODEL

The forward diffusion process with $T$ iterations of a DDPM model is defined as a fixed posterior distribution $p(\mathbf{x}_{1:T}|\mathbf{x}_0)$. Given a list of fixed variance schedule $\beta_1, ..., \beta_T$, we can define a Markov

chain process:

$$p(\mathbf{x}_{1:T}|\mathbf{x}_0) = \prod_{t=1}^{T} p(\mathbf{x_t}|\mathbf{x_{t-1}})$$

$$p(\mathbf{x}_t|\mathbf{x}_{t-1}) = \mathcal{N}(\mathbf{x}_t; \sqrt{1-\beta_t}\mathbf{x}_{t-1}, \beta_t I). \tag{31}$$

We have the following property:

**Property 1.** *The marginal distribution of the forward diffusion process $p(\mathbf{x}_t|\mathbf{x}_0)$ can be written as:*

$$p(\mathbf{x}_t|\mathbf{x}_0) = \mathcal{N}(\mathbf{x}_t; \sqrt{\bar{\alpha}_t}\mathbf{x}_0, (1-\bar{\alpha}_t)I). \tag{32}$$

This can be obtained by the following proof:

*Proof.* Using $p(\mathbf{x}_t|\mathbf{x}_{t-1})$ from equation 31, we can obtain:

$$\begin{aligned}
\mathbf{x}_t &= \sqrt{\alpha_t}\mathbf{x}_{t-1} + \sqrt{\beta_t}\mathbf{z}_t \\
&= \sqrt{\alpha_t\alpha_{t-1}}\mathbf{x}_{t-1} + \sqrt{\alpha_t\beta_{t-1}}\mathbf{z}_{t-1} + \sqrt{\beta_t}\mathbf{z}_t \\
&= ... \\
&= \sqrt{\bar{\alpha}_t}\mathbf{x}_0 + \sqrt{\alpha_t\alpha_{t-1}...\alpha_2\beta_1}\mathbf{z}_1 + ... + \sqrt{\alpha_t\beta_{t-1}}\mathbf{z}_{t-1} + \sqrt{\beta_t}\mathbf{z}_t.
\end{aligned} \tag{33}$$

We can see that $p(\mathbf{x}_t|\mathbf{x}_0)$ can be written as a Gaussian with mean $\sqrt{\bar{\alpha}_t}\mathbf{x}_0$ and variance $(\alpha_t\alpha_{t-1}...\alpha_2\beta_1 + ... + \alpha_t\beta_{t-1} + \beta_t)I = (1-\bar{\alpha}_t)I$. □

This property allows us to write the forward diffusion process in the form of equation 5. As $T \to \infty$, the discretized equation 5 converges to the SDE form defined in equation 4.

**Lemma 1.** *(Tweedie's formula) Let $\mu$ be sampled from a prior probability distribution $G(\mu)$ and $z \sim \mathcal{N}(\mu, \sigma^2)$, the posterior expectation of $\mu$ given $z$ is as:*

$$\mathbb{E}[\mu \mid z] = z + \sigma^2 \nabla_z \log p(z). \tag{34}$$

From Tweedie's formula, we can obtain the following property:

**Property 2.** *For DDPM with the marginal distribution $p(\mathbf{x}_t|\mathbf{x}_0)$ of the forward diffusion process computed in equation 32, $p(\mathbf{x}_0|\mathbf{x}_t)$ has a posterior mean at:*

$$\mathbb{E}[\mathbf{x}_0 \mid \mathbf{x}_t] = \frac{1}{\sqrt{\bar{\alpha}(t)}}(\mathbf{x}_t + (1-\bar{\alpha}(t))\nabla_{\mathbf{x}_t} \log p_t(\mathbf{x}_t)). \tag{35}$$

## C  ALGORITHMS

### C.1  TRAINING AND SAMPLING ALGORITHM OF BACKDIFF

We provide the training procedure in Algorithm 1 and the manifold constraint sampling procedure in Algorithm 2.

### C.2  CG ATOMS CHOICE STRATEGIES

We elaborate on the CG atoms' choice strategies for the self-supervised training framework, as described in Sec. 4.2. The random strategy is shown in Algorithm 3 and the semi-random strategy is shown in Algorithm 4. In this work, we choose a semi-random strategy throughout the training, with the training ratio defined in Table 4. The training ratio value is obtained by roughly estimating the usage of each atom type in popular CG models. We notice that incorporating a larger percentage of other atom types not listed, while enhancing the generalization across different CG protocols, will require longer training time. Except for the training ratio of $C_\alpha$, users can adjust the other values as needed.

---

**Algorithm 1** Training of Backdiff

---

1: **Input:** proteins $[\mathcal{G}_0, ..., \mathcal{G}_N]$, each with ensembles $[\mathcal{C}_0, ..., \mathcal{C}_{K_\mathcal{G}}]$, learning rate $a$, CG choice strategy $\mathcal{T}$, sequence of noise levels $[\alpha_1, ..., \alpha_T]$
2: **Output:** trained score model $\mathbf{s}_\theta$
3: **for** $i = 1$ to $N_{\text{iter}}$ **do**
4:     **for** $\mathcal{G} \sim [\mathcal{G}_0, ..., \mathcal{G}_N]$ **do**
5:         uniformly sample $t \sim [1, ..., T]$ and $\mathcal{C} \sim [\mathcal{C}_0, ..., \mathcal{C}_{K_\mathcal{G}}]$
6:         Separate $\mathcal{C}$ into CG atoms $\mathcal{R}_{\text{atm}}$ and omit atoms (backmapping targets) $\mathcal{C}_{\text{omit}}$ by the CG choice strategy $\mathcal{T}$ with the observation mask $\mathcal{M}$
7:         Calculate the displacement $\mathcal{D}$ of each omitted atom from its residue's $C_\alpha$'s position
8:         $\mathbf{z} \sim \mathcal{N}(0, I)$
9:         Calculate noisy displacement $\mathcal{D}_t = \sqrt{\alpha_t}\mathcal{D} + (1 - \alpha_t)\mathbf{z}$
10:         Obtain noisy configuration $\mathcal{C}_t$ from $\mathcal{D}_t$
11:         predict $\hat{\mathbf{s}} = \mathbf{s}_{\theta,\mathcal{G}}(\mathcal{C}_t, \mathcal{M}, t)$
12:         update $\theta \leftarrow \theta - a\nabla_\theta \left\| \hat{\mathbf{s}} - \nabla_{\mathcal{D}_t} \log p_{t|0}(\mathcal{D}_t \mid \mathbf{0}) \right\|^2$
13:     **end for**
14: **end for**

---

**Algorithm 2** BackDiff sampling with manifold constraint

---

1: **Input:** protein molecular graph $\mathcal{G}$, CG mask $\mathcal{M}$, diffusion steps $T$, CG atoms $\mathcal{R}_{\text{atm}}$, CG auxiliary variables $\mathcal{R}_{\text{aux}}$, auxiliary CG mapping function $\xi_{\text{aux}}$, $\{\zeta_i\}_{i=1}^T$, $\{\tilde{\sigma}_i\}_{i=1}^T$, sequence of noise levels $[\alpha_1, ..., \alpha_T]$
2: **Output:** predicted conformers $\mathcal{C}$
3: $\mathcal{D}_t \sim \mathcal{N}(\mathbf{0}, \boldsymbol{I})$
4: **for** $i = T - 1$ to $0$ **do**
5:     Obtain noisy configuration $\mathcal{C}_i$ from $\mathcal{D}_i$ and $\mathcal{R}_{\text{atm}}$
6:     $\hat{\mathbf{s}} \leftarrow \mathbf{s}_\theta(\mathcal{C}_i, \mathcal{M}, t)$
7:     $\hat{\mathcal{D}}_0 \leftarrow \frac{1}{\sqrt{\bar{\alpha}_i}}(\mathcal{D}_i + (1 - \bar{\alpha}_i)\hat{\mathbf{s}})$
8:     $\boldsymbol{z} \sim \mathcal{N}(\mathbf{0}, \boldsymbol{I})$
9:     $\mathcal{D}'_{i-1} \leftarrow \frac{\sqrt{\alpha_i}(1 - \bar{\alpha}_{i-1})}{1 - \bar{\alpha}_i}\mathcal{D}_i + \frac{\sqrt{\bar{\alpha}_{i-1}}\beta_i}{1 - \bar{\alpha}_i}\hat{\mathcal{D}}_0 + \tilde{\sigma}_i\boldsymbol{z}$
10:     $\mathcal{D}_{i-1} \leftarrow \mathcal{D}'_{i-1} - \zeta_i\nabla_{\mathcal{D}_i}\left\| \mathcal{R}_{\text{aux}} - \xi_{\text{aux}}\left(\hat{\mathcal{D}}_0, \mathcal{R}_{\text{atm}}\right) \right\|_2^2$
11: **end for**
12: Obtain $\mathcal{C}$ from $\hat{\mathcal{D}}_0$ and $\mathcal{R}_{\text{atm}}$

---

|   | $C_\alpha$ | $N$ | $C$ | $O$ | $C_\beta$ | Other |
|---|---|---|---|---|---|---|
| r | 1 | 0.6 | 0.6 | 0.4 | 0.4 | 0.05 |

Table 4: The training ratio of each atom type. Atoms with the same atom types will have the same training ratio.

## D  CG MAPPING PROTOCOLS

In this section, we briefly introduce the three CG methods used for backmapping experiments in this paper. These CG models are designed from a mixing of knowledge-based and physics-based potentials and have been successfully applied in studying ab initio protein structure prediction, protein folding and binding, and extended to even larger systems like protein-DNA interactions. The CG mapping protocol of each method will vary from systems. In this paper, we take the general form of each protocol, summarized in Table 5. Among the three chosen CG methods, MARTINI has the highest CG resolutions: roughly four sidechain heavy atoms represented by one CG atom and two heavy atoms on the ring-like structure represented by one CG atom.

---

**Algorithm 3** CG atoms choice: random strategy

---

1: **Input:** a training sample with N heavy atoms: $\mathcal{C} = [\boldsymbol{c}_1, \boldsymbol{c}_2, \cdots, \boldsymbol{c}_N]$
2: **Output:** CG atoms $\mathcal{R}_{\text{atm}}$, omitted atoms $\mathcal{C}_{\text{omit}}$, CG mask $\mathcal{M} = [m_1, m_2, \cdots, m_N]$
3: CG atom ratio $r \sim \text{Uniform}(0, 1)$
4: **for** $i = 1$ to N **do**:
5:  **if** atom $i$ is a $C_\alpha$ **then**
6:   $\mathcal{C}_i \in \mathcal{R}_{\text{atm}}$
7:   $m_i = 0$
8:  **else**
9:   $p_i \sim \text{Uniform}(0, 1)$
10:   **if** $p_i > r$ **then**
11:    $\mathcal{C}_i \in \mathcal{C}_{\text{omit}}$
12:    $m_i = 1$
13:   **else**
14:    $\mathcal{C}_i \in \mathcal{R}_{\text{atm}}$
15:    $m_i = 0$
16:   **end if**
17:  **end if**
18: **end for**

---

**Algorithm 4** CG atoms choice: semi-random strategy

---

1: **Input:** a training sample with N heavy atoms: $\mathcal{C} = [\boldsymbol{c}_1, \boldsymbol{c}_2, \cdots, \boldsymbol{c}_N]$, a pre-defined training ratio $r = [r_1, r_2, ..., r_N]$
2: **Output:** CG atoms $\mathcal{R}_{\text{atm}}$, omitted atoms $\mathcal{C}_{\text{omit}}$, CG mask $\mathcal{M} = [m_1, m_2, \cdots, m_N]$
3: CG atom ratio $r \sim \text{Uniform}(0, 1)$
4: **for** $i = 1$ to N **do**:
5:  $p_i \sim \text{Uniform}(0, 1)$
6:  **if** $p_i > r_i$ **then**
7:   $\mathcal{C}_i \in \mathcal{C}_{\text{omit}}$
8:   $m_i = 1$
9:  **else**
10:   $\mathcal{C}_i \in \mathcal{R}_{\text{atm}}$
11:   $m_i = 0$
12:  **end if**
13: **end for**

---

| | $R_{\text{atm}}$ | $R_{\text{aux}}$ |
|---|---|---|
| MARTINI | $C_\alpha$ | Up to Four side chain COM beads |
| Rosetta | $C_\alpha, C, N, O$ | side chain COM |
| UNRES | $C_\alpha, N$ | side chain COM |

Table 5: The CG mapping protocol of three CG methods used in this paper.

## E    MODIFIED TORSIONAL DIFFUSION

In this section, we briefly introduce Torsional Diffusion. Torsional Diffusion is a diffusion framework operating on the space of torsion angles. Torsion angles describe the rotation of bonds within a molecule. It lies in $[0, 2\pi)$, and a set of $m$ torsion angles define a hypertorous space $\mathbb{T}^m$. The theory behind score-based diffusion holds for compact Riemannian manifolds, with subtle modifications. For $\boldsymbol{\tau} \in M$, where $\boldsymbol{\tau}$ represents the torsion angles and $M$ is Riemannian manifold, the prior distribution $p_T(\mathbf{x})$ is a uniform distribution over $M$. We choose VE-SDE as our forward diffusion, with $\mathbf{f}(\boldsymbol{\tau}, t) = 0, g(t) = \sqrt{\frac{d}{dt}\sigma^2(t)}$, where $\sigma(t)$ represents the noise scale. We use an exponential diffusion $\sigma(t) = \sigma_{\min}^{1-t}\sigma_{\max}^t$, with $\sigma_{\min} = 0.01\pi, \sigma_{\max} = \pi, t \in (0, 1)$. As shown in equation 3, training a denoising score matching model requires sampling from the perturbation kernel $p(\boldsymbol{\tau}(t)|\boldsymbol{\tau}(0))$. We consider the perturbation kernel on $\mathbb{T}^m$ with wrapped normal distribution:

$$p(\boldsymbol{\tau}(t)|\boldsymbol{\tau}(0)) \propto \sum_{\mathbf{d}\in\mathbb{Z}^m} \exp\left(-\frac{\|\boldsymbol{\tau}(0) - \boldsymbol{\tau}(t) + 2\pi\mathbf{d}\|^2}{2\sigma^2(t)}\right), \tag{36}$$

and the other terms in the loss function equation 3 remain unchanged.

The sampling process of Torsional Diffusion is also similar to normal diffusion models with little changes: we draw samples from a uniform distribution as prior on torus space, and then discretize and solve the reverse diffusion via a geodesic random walk. We implement the model as a Torsional Diffusion conditioned on CG variables. The sampling procedure of the modified Torsional Diffusion is shown in the pseudo-code in Algorithm. 5.

---

**Algorithm 5** Modified Torsional Diffusion sampling

1: **Input:** protein molecular graph $\mathcal{G}$, diffusion steps $T$, CG atoms $\mathcal{R}_{\text{atm}}$, auxiliary variables $\mathcal{R}_{\text{aux}}$ (including bond lengths $l$ and bond angles $\omega$)
2: **Output:** predicted conformers $\mathcal{C}$
3: $\boldsymbol{\tau}_T \sim U(0, 2\pi)^m$
4: **for** $i = T - 1$ to $0$ **do**
5:     let $t = i/T, g(t) = \sigma_{\min}^{1-t}\sigma_{\max}^t \sqrt{2\ln(\sigma_{\max}/\sigma_{\min})}$
6:     Obtain noisy configuration $\mathcal{C}_i$ from $\boldsymbol{\tau}_i, \mathcal{R}_{\text{atm}}, l, \omega$
7:     $\hat{\mathbf{s}} \leftarrow \mathbf{s}_{\theta,\mathcal{G}}(\mathcal{C}_i, t)$
8:     $\mathbf{z} \sim$ wrapped normal with $\sigma^2 = 1/T$
9:     $\boldsymbol{\tau}'_{i-1} = \boldsymbol{\tau}_i + (g^2(t)/N)\hat{\mathbf{s}}$
10:    $\boldsymbol{\tau}_{i-1} = \boldsymbol{\tau}'_{i-1} + g(t)\mathbf{z}$
11: **end for**
12: Obtain $\mathcal{C}$ from $\boldsymbol{\tau}'_0, \mathcal{R}_{\text{atm}}, l, \omega$

---

## F    EVALUATION METRICS

**Root Mean Squared Distance (RMSD)** Root Mean Square Deviation (RMSD) is a commonly used measure in structural biology to quantify the difference between two protein structures. It's particularly useful for comparing the similarity of protein three-dimensional structures. The RMSD is calculated by taking the square root of the average of the square of the distances between the

atoms of two superimposed proteins:

$$\text{RMSD} = \min_{T_g \in \text{SE}(3)} \sqrt{\frac{1}{N} \sum_{i=1}^{N} \left\| T_g(\mathbf{r}_i) - \mathbf{r}_i^{\text{ref}} \right\|^2} \tag{37}$$

, where $N$ is the number of atoms in the protein, and $\mathbf{r}_i$ and $\mathbf{r}_i^{\text{ref}}$ are positions of the $i-$th equivalent atoms of two structures being compared. A lower RMSD of a generated configuration indicates more similarity to the original all-atom configuration.

**Generative diversity score (DIV)** RMSD can be a confusing metric when evaluating the diversity of the generated samples. The main reason lies in that a high RMSD can simultaneously indicate high diversity and low accuracy. As suggested by Jones et al. (2023), the average pairwise RMSDs between (1) generated samples and the original reference and (2) between all generated samples should be approximately equal. Following this idea, a generative diversity score DIV is defined as:

$$\text{RMSD}_{\text{ref}} = \frac{1}{N} \sum_{i}^{N} \text{RMSD}\left(\mathcal{C}_i^{\text{gen}}, \mathcal{C}^{\text{ref}}\right)$$

$$\text{RMSD}_{\text{gen}} = \frac{2}{N(N-1)} \sum_{i}^{N} \sum_{j}^{(i-1)} \text{RMSD}\left(\mathcal{C}_i^{\text{gen}}, \mathcal{C}_j^{\text{gen}}\right) \tag{38}$$

$$\text{DIV} = 1 - \frac{\text{RMSD}_{\text{gen}}}{\text{RMSD}_{\text{ref}}},$$

where N is the number of generated samples conditioned on a single CG configuration. DIV approximately lies on the interval $[0, 1]$. A deterministic backmapping (all generated samples are the same) will have DIV $= 1$, indicating no diversity. On the contrary, DIV $\approx 0$ is achieved when $\text{RMSD}_{\text{ref}} \approx \text{RMSD}_{\text{gen}}$, which indicates $\mathcal{C}^{\text{ref}}$ and $\mathcal{C}_i^{\text{gen}}$ shares a similiar distribution. In this case, the backmapping algorithm generates diverse all-atom configurations following a correct probability distribution. Overall this metric can indicate diversity well and avoid giving high diversity scores (low DIV) to models that generate totally off configurations.

**Steric clash ratio** A steric clash in protein structures refers to a situation where atoms are positioned too close to each other, leading to overlapping electron clouds. This results in an energetically unfavorable state, as it violates the principles of van der Waals radii and can destabilize the protein structure. Following GenZProt (Yang & Gómez-Bombarelli (2023)), we report the ratio of steric clash occurrence in all atom-atom pairs within 5.0 Å, where the steric clash is defined as an atom-atom pair with a distance smaller than 1.2 Å.

## G EXPERIMENT DETAILS

### G.1 MODEL ARCHITECTURE

Graph Neural Network (GNN) has been widely applied in molecular conformation prediction problems. In this paper, we adopt the equivariant GNN, and more specifically, e3nn library as our GNN architecture to parameterize the conditional score function $\mathbf{s}_\theta$. Following Batzner et al. (2022), we denote each node $a$ with node representations $V_{acm}^{k,l,p}$, where $k$ represents the message-passing layer number, $l$ represents the rotation order, $p \in [-1, 1]$ represents the parity, with $p = 1$ representing even parity (invariant under parity), and $p = -1$ representing odd parity (equivariant under parity).

In this study, we denote the choice of CG atoms with an observation mask $\mathcal{M} = \{n_1, ..., n_N\} \in \{0, 1\}^N$, with $n_a = 0$ representing the $a$-th atom is a CG atom and $n_a = 1$ representing the $a$-th atom is an omitted atom. We then have each protein configuration data input expressed as $\{\mathcal{D}, \mathcal{R}_{\text{atm}}, \mathcal{M}, \mathcal{G}\}$. Each node in the graph $\mathcal{G}$ is represented as $v_a = \{n_a, t_a\}$, where $n_a$ is a learnable atom type embedding fixed for a given atomic number and $t_a$ is a learnable amino acid type embedding fixed for a given amino acid. Each edge in the graph is represented as $e_{ab} = \{v_a + v_b, s_{ab}, \mu(d_{ab}), t_{\text{GRF}}\}$, where $s_{ab}$ is a learnable bond type embedding for a given bond type, $\mu(d_{ab})$ is the radial basis representation of distance between node $a$ and node $b$, and $t_{\text{GRF}} = \{\sin 2\pi\omega t, \cos 2\pi\omega t\}$ represents the diffusion time information with Gaussian random features. Given the protein configuration data input $\{\mathcal{D}, \mathcal{R}_{\text{atm}}, \mathcal{M}, \mathcal{G}\}$ and the diffusion time $t$, we first

embed node and edge attributes into higher dimensional feature spaces using feedforward networks:

$$V_a^{0,0,1} = \text{MLP}(v_a) \quad \forall v_a \in \mathcal{V},$$
$$\mathbf{h}_{e_{ab}} = \text{MLP}(e_{ab}) \quad \forall e_{ab} \in \mathcal{E}. \tag{39}$$

The message-passing layers are based on E(3) equivariant convolution from Batzner et al. (2022), Jing et al. (2022). At each layer, messages passing between two paired nodes are constructed using tensor products of nodes' irreducible representation with the spherical harmonic of edge vectors. The messages are weighted by a learnable function that takes in the scalar representations ($l = 0$) of two nodes and edges. Finally, the tensor product is computed via contract with the Clebsch-Gordan coefficients. At the message-passing layer $k$, for the node $a$, its rotation order $l_0$, and output dimension $c'$, the message-passing layer is expressed as:

$$V_{ac'm_o}^{(k,l_o,p_o)} = \sum_{l_f,l_i,p_i} \sum_{m_f,m_i} C_{(l_i,m_i)(l_f,m_f)}^{(l_o,m_o)} \frac{1}{|\mathcal{N}_a|} \sum_{b\in\mathcal{N}_a} \sum_c \psi_{abc}^{(k,l_o,l_f,l_i,p_i)} Y_{m_f}^{(l_f)}(\hat{r}_{ab}) V_{bcm_i}^{(k-1,l_i,p_i)}, \tag{40}$$

where the tensor product between the input feature of rotation order $l_i$ and spherical harmonics of order $l_f$ generates irreducible representations of output orders $|l_i - l_f| \leq l_o \leq |l_i + l_f|$, $(-1)^{l_f} p_i = p_o$, $C$ represents the Clebsch-Gordan coefficients, $\mathcal{N}_a = \{b \mid \forall e_{ab} \in \mathcal{E}\}$ represents the neighboring nodes of node $a$, $Y$ represents the spherical harmonics, and

$$\psi_{abc}^{(k,l_o,l_f,l_i,p_i)} = \Psi_c^{(k,l_o,l_f,l_i,p_i)}\left(\mathbf{h}_{e_{ab}} \left\| V_a^{(k-1,0,1)} \right\| V_b^{(k-1,0,1)}\right) \tag{41}$$

is the weight function using feedforward networks that take in the scalar representations of two nodes and the edge embeddings. In this paper, the rotational order of nodes $(l_0, l_i)$ and spherical harmonics $(l_f)$ are below 3.

After $L$ layers of message-passing, the node feature becomes $V_a = (V^{(L,0,p)} \in \mathbb{R}^c, V^{(L,1,p)} \in \mathbb{R}^{3c}, V^{(L,2,p)} \in \mathbb{R}^{5c})$. We parameterize the time-dependent score function $\mathbf{s}_\theta(\mathcal{D}(t), t|\mathcal{R}_{\text{atm}})$ with rotational and parity equivariant feature $V_a^{(L,1,-1)}$:

$$\mathbf{s}_\theta = [V_a^{(L,1,-1)} : n_a = 1]. \tag{42}$$

## G.2 Hyperparameters

In this section, we introduce the details of our experiments. The score function $\mathbf{s}_\theta$ is parameterized by the equivariant GNN presented in Sec. G.1. The atom type embedding $n_a$ has an embedding size of 4 and the amino acid type embedding $t_a$ has an embedding size of 8. The bond type embedding $s_{ab}$, which denotes if an edge represents a bonded or nonbonded interaction, has an embedding size of 2. In the initial embedding step, node and edge features are embedded into a latent dimension of 32. 8 message-passing layers as in equation 40 are used. The final 3-dimensional rotational and parity equivariant output features of each omitted atom are concatenated as the final predicted score. For the hyperparameters of the VP-SDE, we choose $\beta_1 = 1.0 \times 10^{-7}$, $\beta_T = 1.0 \times 10^{-3}$, with a sigmoid $\beta$ scheduler and diffusion step numbers $T = 10000$. BackDiff is trained on a single NVIDIA-A10 GPU until convergence, with a training time of around 24 hours and ADAM as the optimizer, with 64 batch size.

## G.3 Choice of the correction weight

An important hyperparameter in the manifold constraint sampling is the correction term weight $\zeta$. We should expect that a too-low weight will lead to inconsistency with the conditions and an overly-high weight will make the sampling path noisy. Following Chung et al. (2022), we set $\zeta_i = \zeta_i' / \left\| \mathcal{R}_{\text{aux}} - \xi_{\text{aux}}\left(\hat{\mathcal{D}}_0, \mathcal{R}_{\text{atm}}\right) \right\|$, with $\zeta_i' = 0.5$ yielding the optimized sampling quality. An ablation study on the influence of correction weight is summarized in Table 6. From the table, we can see that the proposed correction weights produce the best result. Although a higher correction weight can offer stronger manifold constraints, leading to a smaller bond length and bond angle error, it over-deviates the sampling path and thus generates samples at low probability space.

|  | $\zeta_i'$ | PED00011 | PED00055 | PED00151 |
|---|---|---|---|---|
| Bond length MAE (Å) | 0.5 | **< 0.001** | **< 0.001** | **< 0.001** |
|  | 0.01 | 0.003(< 0.001) | 0.007(0.002) | 0.004(0.001) |
|  | 500 | **< 0.001** | **< 0.001** | **< 0.001** |
| Bond angle MAE | 0.5 | 0.167(0.095) | 0.106(0.088) | 0.124(0.097) |
|  | 0.01 | 0.293(0.164) | 0.176(0.150) | 0.194(0.123) |
|  | 500 | **0.099(0.003)** | **0.078(0.004)** | **0.065(0.002)** |
| SCR (%) | 0.5 | **0.918(0.609)** | **0.786(0.335)** | **0.820(0.316)** |
|  | 0.01 | 2.485(0.743) | 2.201(0.469) | 2.093(0.554) |
|  | 500 | 1.966(0.451) | 1.835(0.644) | 1.752(0.340) |

Table 6: Ablation study on different correction weights.

## H   ADDITIONAL EXPERIMENTAL RESULTS

We present the multi-protein backmapping results for Rosetta CG model in Table 7 and MARTINI CG model in Table 8. Note that the CG-transferable BackDiff model is not retrained for the two new experiments. The results further demonstrate BackDiff's enhanced accuracy and transferability. Notably, in the experiments with the MARTINI CG model, which features a higher dimensionality of CG auxiliary variables, BackDiff achieves superior backmapping results compared to its performance with the other two CG models (UNRES and Rosetta). On the other hand, baseline methods like GenZProt and Torsional Diffusion deliver similar or less impressive results with the MARTINI CG model than with UNRES and Rosetta. This indicates that BackDiff can harness the benefits of CG models with a richer set of auxiliary variables, a capability not apparent in the other methods. Additionally, we evaluate the sidechain torsion angle distribution of ground truth and sampled configurations from different methods. As shown in Figure 3, 4 and 5, BackDiff aligns closer to the ground truth distributions, even though torsion angles aren't its primary training objective.

|  | Method | PED00011 | PED00055 | PED00151 |
|---|---|---|---|---|
| RMSD$_{min}$(Å) | **BackDiff (fixed)** | **0.616(0.201)** | 1.587(0.359) | **1.287(0.163)** |
|  | BackDiff (trans) | 0.751(0.222) | **1.344(0.275)** | 1.410(0.197) |
|  | GenZProt | 2.245(0.430) | 2.568(0.496) | 2.661(0.325) |
|  | TD | 1.599(0.357) | 2.003(0.376) | 1.458(0.256) |
| SCR (%) | **BackDiff (fixed)** | **0.611(0.456)** | **0.784(0.529)** | **0.463(0.268)** |
|  | BackDiff (trans) | 0.923(0.647) | 0.792(0.475) | 0.820(0.316) |
|  | GenZProt | 2.215(1.237) | 2.192(0.673) | 1.545(0.602) |
|  | TD | 1.034(0.499) | 1.205(0.471) | 0.772(0.315) |
| SCMSE$_{min}$ (Å$^2$) | **BackDiff (fixed)** | **0.068(0.010)** | **0.097(0.020)** | 0.119(0.015) |
|  | BackDiff (trans) | 0.075(0.023) | 0.104(0.021) | **0.111(0.044)** |
|  | GenZProt | 1.787(0.289) | 1.704(0.368) | 1.633(0.301) |
|  | TD | 1.108(0.309) | 0.946(0.247) | 1.513(0.350) |
| DIV | BackDiff (fixed) | 0.139(0.056) | 0.261(0.115) | 0.200(0.079) |
|  | **BackDiff (trans)** | **0.084(0.060)** | **0.155(0.067)** | **0.108(0.058)** |
|  | GenZProt | 0.625(0.117) | 0.637(0.132) | 0.604(0.136) |
|  | TD | 0.184(0.061) | 0.271(0.091) | 0.205(0.081) |

Table 7: Results on multi-protein experiments backmapping from Rosetta CG model.

| | Method | PED00011 | PED00055 | PED00151 |
|---|---|---|---|---|
| $\text{RMSD}_{min}(\text{Å})$ | **BackDiff (fixed)** | **0.415(0.156)** | 1.012(0.208) | **0.827(0.141)** |
| | BackDiff (trans) | 0.517(0.182) | **0.827(0.174)** | 0.957(0.196) |
| | GenZProt | 2.993(0.526) | 3.015(0.728) | 2.982(0.552) |
| | TD | 1.969(0.527) | 2.493(0.643) | 1.738(0.216) |
| SCR (%) | **BackDiff (fixed)** | **0.314(0.232)** | **0.629(0.512)** | **0.227(0.135)** |
| | BackDiff (trans) | 0.536(0.478) | 0.701(0.435) | 0.520(0.393) |
| | GenZProt | 2.759(0.988) | 3.000(0.672) | 1.894(0.433) |
| | TD | 1.103(0.570) | 1.741(0.513) | 1.450(0.513) |
| $\text{SCMSE}_{min} (\text{Å}^2)$ | **BackDiff (fixed)** | 0.035(0.008) | **0.030(0.005)** | **0.028(0.005)** |
| | BackDiff (trans) | **0.030(0.006)** | 0.034(0.007) | 0.040(0.015) |
| | GenZProt | 2.307(0.378) | 2.145(0.488) | 2.389(0.404) |
| | TD | 1.302(0.284) | 1.436(0.527) | 1.784(0.496) |
| DIV | BackDiff (fixed) | 0.205(0.050) | 0.325(0.087) | 0.198(0.074) |
| | **BackDiff (trans)** | **0.147(0.072)** | **0.169(0.078)** | **0.152(0.063)** |
| | GenZProt | 0.674(0.130) | 0.691(0.115) | 0.640(0.128) |
| | TD | 0.282(0.056) | 0.326(0.075) | 0.233(0.059) |

Table 8: Results on multi-protein experiments backmapping from MARTINI CG model.

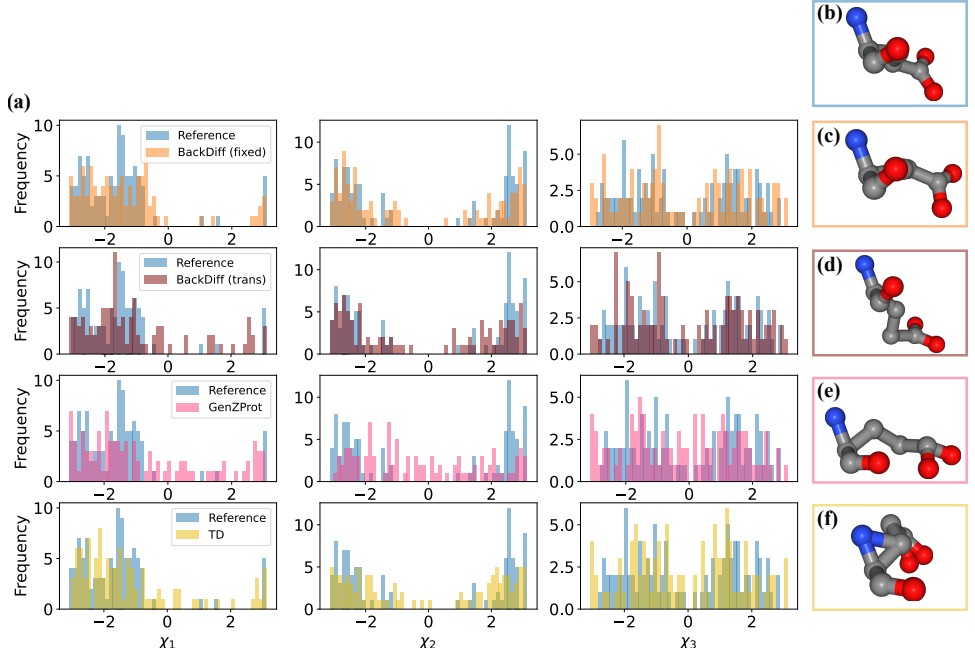

Figure 3: Multi-protein experiments backmapping from the UNRES CG model showing results on residue 7 of PED00011, a Glutamine (GLU) amino acid residue: (a) Histogram of sidechain torsion angles of ground truth and samples generated from four models, (b)-(f): the sidechain configurations visualization from (b) reference (c) fixed CG BackDiff (d) transferable CG BackDiff (e) GenZProt (f) Torsional Diffusion.

# I LIMITATIONS OF BACKDIFF

As shown in Sec. 5, BackDiff significantly improves the protein backmapping accuracy. However, BackDiff has a number of limitations.

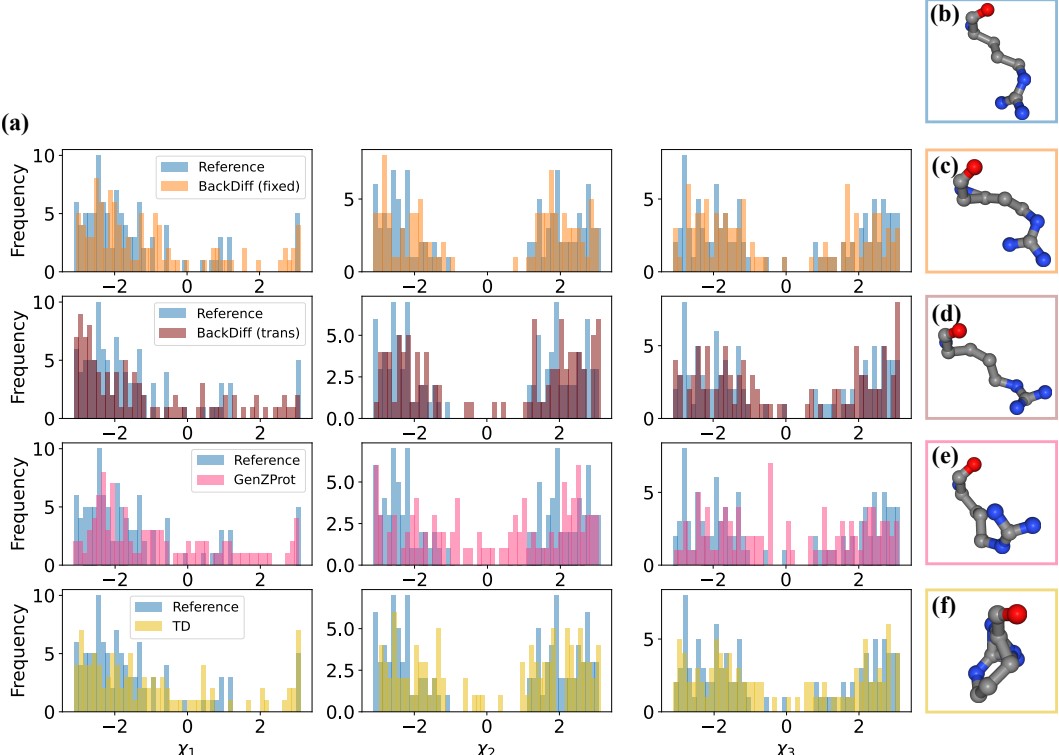

Figure 4: Multi-protein experiments backmapping from the UNRES CG model showing results on residue 8 of PED00011, an Arginine (ARG) amino acid residue.

**Bond lengths and bond angles** A primary drawback of BackDiff, in comparison to internal-coordinate-based generative models, is its susceptibility to producing unrealistic bond lengths and angles, even with manifold constraint sampling. This inaccuracy is notably prominent in bond angles possibly because of their nonlinear mappings from Cartesian coordinates. On the other hand, internal-coordinate-based models inherently avoid such issues by constructing geometries from pre-defined, reasonable bond lengths and angles. Future work will focus on refining these nonlinear manifold constraints to reduce errors in bond angles and other nonlinear CG auxiliary variables.

**Sampling efficiency** A notable limitation of diffusion models is their slower sampling efficiency. Compared to other generative models like Variational Autoencoders (VAE) and Normalizing Flows (NF), which often achieve generation in a single step, diffusion models require hundreds to thousands of reverse diffusion steps for effective sampling. This demand is even more pronounced for manifold constraint sampling, where fewer diffusion steps might not sufficiently constrain the conditions. In BackDiff, generating 100 samples per frame requires an average of 293 seconds, whereas for GenZProt (a VAE-based method) takes an average of 0.009 seconds. Improving the sampling efficiency of both diffusion models and manifold constraint sampling presents a compelling direction for future research.

**Training data quality** An optimal training dataset for BackDiff would encompass data from tens of thousands of proteins, all simulated under a unified force field. Such a dataset would ensure comprehensive coverage of the protein space and minimize inconsistencies in data quality. In contrast, our current dataset comprises a mere 92 proteins, sourced from varied simulations and sampling methodologies. Such diversity in data origins may compromise the model's broader applicability across protein spaces. Moving forward, our goal is to integrate a more expansive and consistent set of high-quality protein simulation data, enhancing the robustness and performance of BackDiff.

**Chirality of proteins** Proteins are made up of amino acids, most of which are chiral. This means they exist in two forms (enantiomers) that are mirror images of each other but cannot be superimposed. In nature, almost all amino acids in proteins are in the L-form (left-handed). This chirality is

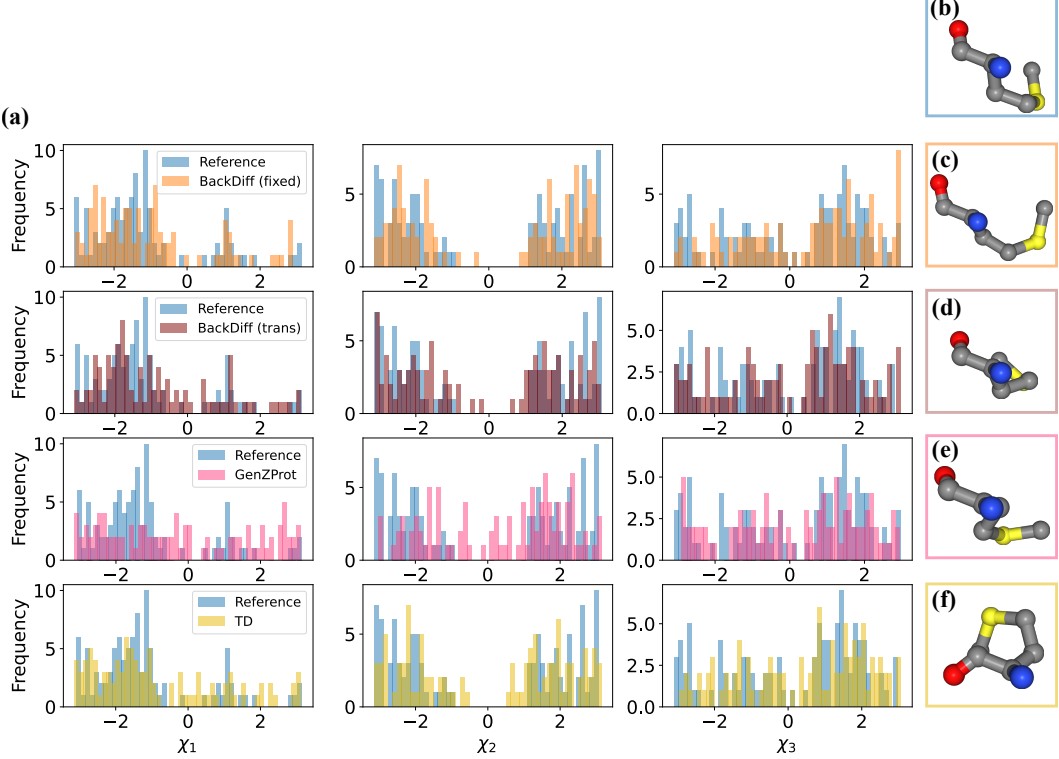

Figure 5: Multi-protein experiments backmapping from the UNRES CG model showing results on residue 27 of PED0001, a Methionine (MET) amino acid residue.

crucial for the structure and function of proteins. Performing a parity transformation on the protein will change left-handed coordinate systems into right-handed ones. Our model does not take care of the chirality and simply assumes parity equivariant: $p(\mathcal{C}|\mathcal{R}_{\text{atm}}, \mathcal{G}) = p(-\mathcal{C}| - \mathcal{R}_{\text{atm}}, \mathcal{G})$. This can be a point for improvement.

