# OpenReview forum: "Backdiff: a diffusion model for generalized transferable protein backmapping"
_ICLR.cc/2024/Conference — Submitted to ICLR 2024_

### Official Review · Reviewer_gYSb · 2023-10-28

**Soundness:** 2 fair
**Presentation:** 2 fair
**Contribution:** 2 fair
**Rating:** 3
**Confidence:** 4

**Summary:**

The manuscript presents a method based on a diffusion model for backmapping coarse-grained MD results to full atom coordinates. By incorporating self-supervised training strategies, the proposed method can be generalized to multiple different coarse-graining (CG) methods. Experimental results indicate that the proposed method achieves better performance than state-of-the-art methods in backmapping CG configurations.

**Strengths:**

1. The proposed method can be applied to multiple different CG methods without the need for retraining.

2. Experimental results demonstrate that the proposed method outperforms baseline methods.

**Weaknesses:**

1. The Equivariance handling approach in the method constructs a reference coordinate system using the first three amino acids of each protein sequence. This implies that if the positions of the first three amino acids vary, the reference system will also differ, which may not be an ideal approach.

2. In Table 3, the Mean Absolute Error (MAE) of bond length for BackDiff (cons) can reach 0, which appears too good to be true.

3. In Table 3, the numerical values of the standard deviation (std) in the second and third rows are almost in the same range as the mean values, which is strange.

**Questions:**

1. Is the model used in the method SE3 equivariant?

2. Given that the model learns the displacement of omitted atoms from alpha carbons, why not directly learn displacement in the local coordinate system of each amino acid? This approach could ensure that the representation is SE3 equivariant.

3. I am not familiar with the PED dataset. Why were only 92 proteins selected out of 227 for training and testing data?

4. What do you mean by single- and multi-protein experiments? What is the primary difference? When frames are used for data partitioning, is it possible for different frames of the same protein to appear in both the training and testing sets, potentially leading to data leakage?

---

> ### Author Response · Authors · 2023-11-17
> **Summary of revised draft**
>
> Dear reviewer,
> I sincerely appreciate the time and effort you invested in reviewing my paper. Your insightful comments and suggestions have been instrumental in enhancing the quality of this work. Before moving on to specific questions, I would like to summarize major revisions made in the new version of the paper, as highlighted in light blue in the paper:
>
> 1. Integration of Equivariant GNN: We have now adapted an equivariant GNN to parameterize the time-dependent score function. We have provided theoretical proof demonstrating how an SE(3)-equivariant score function can effectively lead to the desired equivariant conditional. The revision is made in Section 4.3 (Proposition), Appendix A.2 (Proof), and Appendix G (Model Architecture).
>
> 2. Introduction of new metrics: In response to the need for more comprehensive evaluation criteria, new metrics have been introduced to assess the accuracy and diversity of the generated samples. The new metrics are briefly summarized, which you can find details in Evaluation Metrics in Section 5 and Appendix F:
>
>                * Accuracy: the minimum RMSD of generated structures w.r.t the original all-atom configuration; sidechain COM MAE of the generated structure with the minimum RMSD from the original all-atom configuration.
>                * Diversity: generated diversity score which indicates diversity while avoiding giving high diversity scores (low DIV) to models that generate totally off configurations. This metric is suggested in [1].
>                * Physical realistic: steric clash ratio
>
> 3. Testing data adjustments: We have removed the disordered test protein (PED00025) from our testing data. In its place, we have included a mostly globular protein (PED00055) to provide a more robust and representative test case. The revision can be found in Section 5.
>
> All training and inference are rerun on the same devices with new results and figures reported.
>
> Reference:
> [1] Jones, Michael S., Kirill Shmilovich, and Andrew L. Ferguson. "DiAMoNDBack: Diffusion-Denoising Autoregressive Model for Non-Deterministic Backmapping of Cα Protein Traces." Journal of Chemical Theory and Computation (2023).

---

> ### Author Response · Authors · 2023-11-17
> **Response to each question**
>
> Dear reviewer,
> I want to express my deepest gratitude for the great questions and advice you raise. Below is a detailed response to each question:
> 1. Is the model used in the method SE3 equivariant?
> * Thank you for insightful questions regarding the SE(3) equivariance. Our model in the first draft indeed does not use an equivariant neural network to preserve the SE(3) equivariance. Instead, we built a coordinate framework internally for each CG configuration and we used the framework in the diffusion model. However, as you pointed out, this approach might limit the model's expressivity. Thus, we have re-implemented an SE(3) equivariant BackDiff and have updated all results using the new equivariant model. We also prove that with an equivariant score function, we can achieve the desired SE(3)-equivariance.
> 2. Given that the model learns the displacement of omitted atoms from alpha carbons, why not directly learn displacement in the local coordinate system of each amino acid? This approach could ensure that the representation is SE3 equivariant.
> * Thank you for your excellent advice regarding the use of local coordinate systems for each amino acid to ensure SE(3)-equivariance. This approach presents a valuable alternative to our current method. We appreciate this advice and will explore its feasibility in future research.
> 3. I am not familiar with the PED dataset. Why were only 92 proteins selected out of 227 for training and testing data?
> * I apologize for any confusion regarding the selection of proteins. We select proteins in the PED dataset following the guidelines discussed in the GenZProt paper. Metal ion-binding complexes, nucleotide-binding complexes, cofactor-binding complexes, PTM-including proteins except phosphorylation and oxidation, D-amino acid protein, and proteins simulated or experimentally measured in unnatural conditions are excluded. The 92 proteins selected are those computed from reliable MD simulations and sampling methods.
> 4. What do you mean by single- and multi-protein experiments? What is the primary difference? When frames are used for data partitioning, is it possible for different frames of the same protein to appear in both the training and testing sets, potentially leading to data leakage?
> * We apologize for the unclear presentation in the manuscript. To clarify: single-protein experiments involve using data from a single protein. We partition ensembles of frames from this single protein into training and testing datasets. This tests the model's ability to predict structures within a single protein's conformational space. In contrast, multi-protein experiments involve training and testing the model on multiple different proteins. This tests the model's ability to generalize across various proteins. Due to the large conformation space of long-chain proteins, and the comparatively small numbers of ensembles in our dataset, we believe data leakage is not an issue. Still, to make our results more convincing, in the new training of the single-protein experiments, we have reduced the number of training data.
> 5. In Table 3, the Mean Absolute Error (MAE) of bond length for BackDiff (cons) can reach 0, which appears too good to be true.
> * Thank you for highlighting this point about the Mean Absolute Error (MAE) of bond length in Table 3.  The bond length MAE does not actually reach 0, but should be expressed more rigorously as "<0.001", which I have fixed in the revised draft. This extremely low MAE for bond length is expected.  Applying manifold constraint on score-based diffusion model is similar to applying a restraint potential to certain degrees of freedom in molecular dynamics, restraining the variation on these DOF. The effectiveness of manifold constraint is better for constraining bond length compared to bond angle. This observation is consistent with recent studies [Improving Diffusion Models for Inverse Problems using Manifold Constraints] and [DIFFUSION POSTERIOR SAMPLING FOR GENERAL NOISY INVERSE PROBLEMS] which implied that, empirically, a linear condition mapping (like bond lengths) tends to be more easily matched than a nonlinear condition mapping (like bond angles). Additionally, as shown in Table 6, the choice of manifold constraint weight plays a significant role. A small weight can lead to error in condition, while a large weight might over-deviate the sampling. The weight we choose is optimized by balancing the condition error and the sampling path.
> 6. In Table 3, the numerical values of the standard deviation (std) in the second and third rows are almost in the same range as the mean values, which is strange.
> * Thank you for raising questions regarding the bond angle MAE. Like in Table 1 and 2, we evaluate the bond angle MAE of generated all-atom configurations for each testing frame, and compute the mean and std across all testing frames. Thus, a large std should be expected as they are results from backmapping different CG configurations.

---

### Official Review · Reviewer_np12 · 2023-11-01

**Soundness:** 3 good
**Presentation:** 3 good
**Contribution:** 3 good
**Rating:** 6
**Confidence:** 4

**Summary:**

This paper presents a generalized transferable backmapping method that can be applied to arbitrary CG mapping without the need for retraining. The paper formulates backmapping as an imputation problem, where the model generates C alpha-atom distance vectors conditioned on CG atoms and CG auxiliary variables (aggregated properties of groups of atoms). The model can achieve generalization across different CG mappings by training with (semi)-randomly selected CG atoms and auxiliary variables. The model generates output in Cartesian coordinate space but produces well-constrained bond lengths and angles by imposing manifold constraints. The model is compared to a recent transferable generative modeling work, with experiments conducted following similar settings including the dataset and metrics, as well as a recent all atom conformer generation model.

This paper shows clear novelty and strengths, but I still have some questions regarding the experiments.

**Strengths:**

1.	This is a first backmapping algorithm generalized for arbitrary CG mappings.
2.	The idea of formulating the generalized backmapping problem as an imputation problem is novel and makes a lot of sense.

**Weaknesses:**

-	Have you re-implemented the baseline models (especially CGVAE) to condition them on other CG variables such as N and sidechain COM for the UNRES benchmark? If BackDiff is conditioned on C alpha, N, and side chain COM, while CGVAE is conditioned only on C alpha as in its original paper, it would be hard to tell if the performance difference is coming from the method or the difference in information given to the models. The same applies for MARTINI and Rosetta benchmarks. Alternatively, you could report the performance of BackDiff conditioned on C alpha only, with no CG auxiliary variable constraints on side chain COM.
-	Table 1 and Table 2 report the mean RMSD across 100 sampled structures. However, a large mean RMSD of the backmapped structures could also suggest high diversity among all atom conformations, rather than high error in the structures, since one CG structure can correspond to many all atom conformations. Reporting the minimum RMSD across 100 samples should be a better metric for assessing error.
-	How did you select the PED entries for testing? The three test proteins all look pretty linear and disordered. How does the model perform on a globular protein?
-	Could you report the diversity of the generated structures conditioned on the same CG structure? For example, in the referred baseline [1], the authors reported quantitative metrics for diversity, such as the Earth Mover’s Distance for side chain torsion angles.
-	Could you provide a speed analysis of your method, for example how much time required to backmap a frame, similar to what was done in [1]?
-	It could be interesting to see how the model performance changes as we increase the CG resolution (the number of CG atoms and CG auxiliary variables), especially in terms of the diversity of generated all atom conformations. This could provide insights into the CG system’s entropy. This is not a requirement, but just a curiosity.

**Questions:**

-	Yang & Bombarelli’s model is called GenZProt and not CGVAE.

---

> ### Author Response · Authors · 2023-11-17
> **Summary of revised draft**
>
> Dear reviewer,
> I sincerely appreciate the time and effort you invested in reviewing my paper. Your insightful comments and suggestions have been instrumental in enhancing the quality of this work. Before moving on to specific questions, I would like to summarize major revisions made in the new version of the paper, as highlighted in light blue in the paper:
>
> 1. Integration of Equivariant GNN: We have now adapted an equivariant GNN to parameterize the time-dependent score function. We have provided theoretical proof demonstrating how an SE(3)-equivariant score function can effectively lead to the desired equivariant conditional. The revision is made in Section 4.3 (Proposition), Appendix A.2 (Proof), and Appendix G (Model Architecture).
>
> 2. Introduction of new metrics: In response to the need for more comprehensive evaluation criteria, new metrics have been introduced to assess the accuracy and diversity of the generated samples. The new metrics are briefly summarized, which you can find details in Evaluation Metrics in Section 5 and Appendix F:
>
>                * Accuracy: the minimum RMSD of generated structures w.r.t the original all-atom configuration; sidechain COM MAE of the generated structure with the minimum RMSD from the original all-atom configuration.
>                * Diversity: generated diversity score which indicates diversity while avoiding giving high diversity scores (low DIV) to models that generate totally off configurations. This metric is suggested in [1].
>                * Physical realistic: steric clash ratio
>
> 3. Testing data adjustments: We have removed the disordered test protein (PED00025) from our testing data. In its place, we have included a mostly globular protein (PED00055) to provide a more robust and representative test case. The revision can be found in Section 5.
>
> All training and inference are rerun on the same devices with new results and figures reported.
>
> Reference:
> [1] Jones, Michael S., Kirill Shmilovich, and Andrew L. Ferguson. "DiAMoNDBack: Diffusion-Denoising Autoregressive Model for Non-Deterministic Backmapping of Cα Protein Traces." Journal of Chemical Theory and Computation (2023).

---

> ### Author Response · Authors · 2023-11-17
> **Response to each question**
>
> Dear reviewer,
> I want to express my deepest gratitude for the great questions and advice you raise. Below is a detailed response to each specific question:
> 1. Comment:  Have you re-implemented the baseline models (especially CGVAE) to condition them on other CG variables such as N and sidechain COM for the UNRES benchmark? If BackDiff is conditioned on C alpha, N, and side chain COM, while CGVAE is conditioned only on C alpha as in its original paper, it would be hard to tell if the performance difference is coming from the method or the difference in information given to the models. The same applies for MARTINI and Rosetta benchmarks. Alternatively, you could report the performance of BackDiff conditioned on C alpha only, with no CG auxiliary variable constraints on side chain COM.
> * Response: Thank you for highlighting the importance of a fair comparison in our experiments. We did reimplement the baseline models. I maintained all the original GenZProt architecture, except that the residue-residue message-passing and residue-atom message-passing will contain additional particles corresponding to additional CG beads other than alpha carbons. I think this is a fair comparison, as our method BackDiff, and the baseline GenZProt and modified Torsional Diffusion all utilize the same model architecture (e3nn package with E(3) equivariant convolution as message-passing layers).
>
> 2. Comment:  Table 1 and Table 2 report the mean RMSD across 100 sampled structures. However, a large mean RMSD of the backmapped structures could also suggest high diversity among all-atom conformations, rather than high error in the structures, since one CG structure can correspond to many all-atom conformations. Reporting the minimum RMSD across 100 samples should be a better metric for assessing error.
> * Response:  Thank you for your valuable suggestion regarding the use of RMSD as a metric. I agree that the mean RMSD can indeed be ambiguous as it might reflect both the accuracy and the diversity of the backmapped structures. A single CG structure can correspond to multiple all-atom conformations, and thus, a high mean RMSD might not directly imply a high error.  We have revised our evaluation strategy to the minimum RMSD as per your suggestion.
>
> 3. Comment:  How did you select the PED entries for testing? The three test proteins all look pretty linear and disordered. How does the model perform on a globular protein?
> * Response:  Thank you for your insightful observation regarding our selection of PED entries for testing.  Including only disordered proteins may limit the scope of our evaluation. To address this, we have revised our test set to incorporate a broader range of protein structures. Specifically, we have replaced one of the previously used test proteins (PED00025, a disordered protein) with PED00055, a well-characterized globular protein. This choice was guided by the aim of assessing the model's performance across a more diverse range of protein structures, including those with more defined and compact conformations.
>
> 4. Comment:  Could you report the diversity of the generated structures conditioned on the same CG structure? For example, in the referred baseline [1], the authors reported quantitative metrics for diversity, such as the Earth Mover’s Distance for side chain torsion angles.
> * Response:  Diversity is an important metric to evaluate, and we have included such a measurement (DIV) in the new draft. The result shows the BackDiff trained with transferable CG protocol generates samples with the highest diversity overall.
>
> 5. Comment:  Could you provide a speed analysis of your method, for example how much time required to backmap a frame, similar to what was done in [1]?
> * Response:  Certainly! We have added a brief speed analysis in Appendix I: Sampling Efficiency. In general, the diffusion model is slow in sampling, especially with the manifold constraint technique. BackDiff roughly requires around 5 minutes to generate 100 samples for a single frame.
>
> 6. Comment:  It could be interesting to see how the model performance changes as we increase the CG resolution (the number of CG atoms and CG auxiliary variables), especially in terms of the diversity of generated all-atom conformations. This could provide insights into the CG system’s entropy. This is not a requirement, but just a curiosity.
> * Response:  This is indeed an interesting question to investigate. Although we do not put such a comparison in the text, we can observe Table 2 (multi-protein experiment for UNRES, a CG of lower resolutions) and Table 8 (multi-protein experiment for MARTINI, a CG of higher resolutions).  Backmapping from CG of lower resolutions will generate samples with higher diversity (lower DIV score). We suspect that for BackDiff, since MARTINI utilizes multiple sidechain beads, and all this information is used for manifold constraint, the generated samples all get closer to the original all-atom configuration.

---

### Official Review · Reviewer_pCxU · 2023-12-12

**Soundness:** 3 good
**Presentation:** 3 good
**Contribution:** 3 good
**Rating:** 6
**Confidence:** 4

**Summary:**

The authors developed BackDiff, a diffusion model that generates all-atom protein structures from coarse-grained models. During coarse-graining, atoms that are grouped together are represented via auxiliary variables, and the remainder are denoted CG atoms. To train the diffusion model, the authors perform an imputation task, i.e. the missing atoms are generated from a noise distribution conditioned on the remaining CG atoms. To improve generalizability to various coarse-graining methods, the missing atoms can be chosen randomly or semi-randomly from the all-atom configuration during training. In practice, the authors do not generate the coordinates of the missing atoms, instead choosing to generate their displacements from the corresponding carbon atoms in the C-alpha representation, thus necessitating the requirement that all C-alpha atom coordinates are included in the set of CG atoms.

In order to incorporate information in the auxiliary variables, the authors constrain the reverse diffusion process. A manifold constraint is applied to each diffusion step, correcting the configuration via posterior conditions on auxiliary variables, bond lengths, and bond angles.

The authors compare their method against two other backmapping models, GenZProt and TD, showing superior performance in terms of accuracy, diversity, and plausibility.

**Strengths:**

The self-supervised approach enables training for specific coarse-graining techniques as well as generalizability to data from multiple CG methods.

**Weaknesses:**

I am concerned about potential data leakage, based on random splitting of frames into training, test, and validation sets.

To compute accuracy, the authors first identify a generated sample that has the lowest RMSD to the all-atom reference configuration. They then report the MSE of the center of mass of the side chain atoms compared to the reference (SCMSE). Instead, I’d prefer to see the distribution of RMSD across all generated structures, starting from different coarse-grained inputs. This would be helpful in evaluating different CG strategies.

**Questions:**

In Table 1, why does the BackDiff (fixed) model have significantly lower DIV scores (i.e. more diversity of generated structures) compared to BackDiff (trans)? Intuitively, I’d expect the CG-transferable model to produce more diverse structures.

---

### Meta-Review · Area_Chair_Y8FX · 2023-12-08

**Metareview:**

The paper considers the problem of protein back-mapping, and proposes a self-supervised approach building upon the score-based conditional diffusion model, which can adapt to different coarse-grained models.

The AC carefully read the manuscript and feedback materials. Please note that the late review by reviewer pCxU did not alter the metareview which was conducted before the review was posted.

While the method considers an interesting problem (although somewhat less relevant to the ICLR community than tasks with broader applications such as structure prediction etc. ) and the proposed approach makes sense, there are important concerns remaining which warrant significant revision of the manuscript.

The theoretical results are still problematic : Based on the concerns from Reviewer gYSb on equivariance, the authors revised their model. However, the main text does not provide a clear description of the network definition from which SE(3) equivariance could be derived. Moreover in appendix G.1, an E(3) graph convolutional layer is used without modification, which is problematic for protein structure and automatically violates the authors' claims.

From a computational standpoint: in practice, one might be interested in backmapping thousands of different frames from CG simulation. The computational cost of the proposed approach might be prohibitive due to the use of diffusion backbone for generation.

In terms of experiments, the work uses the PED dataset, similarly to GenZProt (https://arxiv.org/abs/2303.01569). However, two out of four settings used in the GenZProt paper are left out (PED00090 and PED00218). Moreover, the results on RMSD and SCR reported for GenZProt in the present submission cannot be aligned with those in the GenZProt paper, which render the baselines questionable.

Finally, the significance of this work would greatly benefit form adding useful insights, beyond claiming improvement in terms of performance metrics. Among other points, it would be useful to elaborate on the relevance of the single protein setting as it is unclear how it could be chemically transferable.

**Justification For Why Not Higher Score:**

The paper presents theoretical flaws and the experiments are problematic.

**Justification For Why Not Lower Score:**

N/A

---

### Decision · Program_Chairs · 2024-01-16

Reject